# Elevational differences in hydrogeomorphic disturbance regime influence sediment residence times within mountain river corridors

Nicholas A. Sutfin [1,2,3] & Ellen Wohl[1]

High-elevation mountain streams are commonly viewed as erosive environments, but they can retain sediment along river corridors for thousands of years. In 2013, an extreme flood evacuated floodplain sediment in the Colorado Front Range, USA. We use fifty-two [14]C ages collected along four streams prior to the flood to estimate mean residence time of floodplain sediment. Here we show that mountain streams above the elevation of the Pleistocene terminal moraine retain floodplain sediment for longer durations than those at lower elevation, but that wildfires may decrease floodplain sediment residence time at high elevations. Comparison of field sites and differencing of pre- and post-flood lidar show that valley confinement is a significant predictor of residence time, sediment flux, and floodplains disturbed by the 2013 flood. Elevational trends in floodplain disturbance regime also reflect differences in forest type, precipitation pattern, and wildfire regime, which are expected to shift under a changing climate.

[1] Department of Geosciences, Colorado State University, 1482 Campus Delivery Fort Collins, Fort Collins, CO 80523-1482, USA. [2] Integrated Water, Atmosphere, and Ecosystem Education and Research Program, Colorado State University, Fort Collins, CO 80523-1482, USA. [3]Present address: Department of Earth, Environmental, and Planetary Sciences, Case Western Reserve University, Cleveland, OH 44106, USA. Correspondence and requests for materials should be addressed to N.A.S. (email: nicholas.sutfin@case.edu)

E rosion and sedimentation within river corridors is impor-
tant for landscape evolution, natural hazards, sediment
yield, the fate and transport of contaminants[1,2], organic
carbon storage along mountain rivers[3,4], river ecosystems, and
water quality[5]. However, little is known about the duration of
floodplain sediment residence within river corridors so potential
variability in transit times is poorly understood[6].

Previous work shows that floodplain sediment storage can be
influenced by climate[7], but no studies have examined potential
controls on floodplain sediment residence times along a con-
tinuum in high-relief environments. Because valley morphology
regulates localized forces acting on the floodplain and potential
for erosion, the degree of valley confinement may be an impor-
tance influence on floodplain sediment residence time. Valley
confinement, which we quantify as the ratio between channel and
valley width, and sediment storage commonly varies downstream
in mountainous regions[8]. The spatial variability of valley con-
finement largely influences the connectivity between hillslopes
and river channels and the delivery of water and sediment to river
corridors[9,10].

Increased variability and uncertainty in precipitation as a result
of a warming climate are projected to increase the frequency and
magnitude of extreme precipitation events and potential for river
flooding in the western US[11,12], which could greatly influence
erosion and sediment dynamics within river corridors. Con-
temporary events and paleohydrologic reconstructions indicate
that anomalous precipitation and resulting extreme floods sub-
stantially increase riverine transport of sediment[13–16]. Recent
work in the Colorado Rocky Mountains suggests that hillslope
erosion during extreme events can greatly impact floodplains and
may have a significant role in the exhumation rate of mountain
ranges[16,17]. This may be particularly important in high-relief
terrain where elevation boundaries between predominantly
snowmelt and rainfall–runoff could shift[18] and glacial retreat
could expose significant sediment sources to erosion by rivers[19].
Increases in rainfall and decreased glacial extent, for example, are
likely to increase exposure and erosion of unconsolidated gla-
ciated sediments in the Himalayas[20].

The movement of water and sediment down hillslopes can be
greatly impacted by wildfire[21], the severity of which is also
increasing as a result of climate change in coniferous forests of the
western United States[22,23]. Severe wildfire decreases interception
of precipitation by forest canopies and ground cover, inhibits
infiltration of water through soil, results in rapid downslope
delivery of water and an increase in surface erosion, and can
trigger landslides and debris flows[21]. Examination of wildfire
through radiocarbon ages in alluvial fans in the northern Rocky
Mountains has linked climatic and hydrologic conditions with
sediment pulses to river corridors[24–26].

Radiocarbon ages of charcoal from floodplain sediment have
been used to identify periods of sedimentation[27], estimate the
timing of deposition[28], and identify differences in sediment
residence times in small basins (10 km$^2$)[29]. High-resolution esti-
mates of residence times have also been conducted for post-bomb
(1965) radiocarbon analysis, which requires additional data sets
of atmospheric observations following increases in atmospheric
$^{14}$C as a result of nuclear weapons testing[30]. Errors in the
approach of $^{14}$C analysis of charcoal as a proxy for the timing of
sediment deposition include older inbuilt ages from inner wood
or charred dead wood[31], re-working of old charcoal through
heterogeneous bioturbation, erosion, and deposition[28], and
potential for in situ charcoal production that may post-date
underlying fluvially deposited charcoal. We discuss these errors
and pertinent assumptions in the discussion.

The eight study sites in the Colorado Front Range (CFR), USA
span an elevation gradient, the Pleistocene glacial terminal

moraine, and various degrees of valley confinement within river
corridors of a mountain basin (Fig. 1). Distinct forest types with
differences in fire regime and hydrogeomorphic response are
present in the montane zone (~ 1750–2800 m elevation), the
subalpine zone (~ 2800–3400 m elevation), and the alpine zone (
> 3400 m elevation), details for which are provided in the meth-
ods. Because convective thunderstorms in the CFR are more
common in the montane zone[32], these storms typically impact
mountainous areas lower in elevation than alpine and subalpine
zones. As a result, ratios of extreme floods to median flows are
much higher in the CFR montane zone than in the subalpine
zone (~2800–3500 m)[33]. Hydrologic flow modeling in the San
Juan Mountains of Colorado indicates similar elevation thresh-
olds for large floods[34], below which a majority of sediment
appears to be mobilized during extreme events[17,35]. It has been
proposed that a hydroclimatic shift regulates an elevation
threshold for recorded paleofloods that disproportionately impact
river corridors below 2300 m in the CFR[36]. These differences in
hydrogeomorphic disturbance suggest that higher elevations may
have longer residence time of floodplain sediment compared with
lower elevations in the CFR.

We use 52 radiocarbon ages, field surveys, and lidar data sets to
quantify floodplain sediment residence times, examine long-
itudinal trends across an elevation gradient, examine potential
controls on floodplain sediment residence times, and place results
into the context of wildfire. Radiocarbon ages were obtained for
2–15 large charcoal pieces ( > 1 cm$^3$) at each of eight study sites,
and the weighted mean of the calibrated probability distribution
was calculated for each sample, providing proxies for floodplain
sediment ages[37–39]. Site-average values were calculated as the
arithmetic means for all samples at each site and used as proxies
for floodplain sediment residence time. Lidar determination of
floodplain erosion and deposition following an extreme flood in
the study basin provides insight into the localized geomorphic
controls on erosion to complement sediment residence time
estimates.

We find that river corridors at elevations above the Pleistocene
glacial terminal moraine in the North Saint Vrain Creek study
basin (~ 2500 m) exhibit significantly older mean ages of flood-
plain sediment compared with those at lower elevations. Stepwise
multiple linear regression models indicate that valley confinement
is a common predictor of residence time of floodplain sediment,
floodplain sediment flux, and floodplain disturbance during the
extreme flood in 2013. Stream power is also a significant predictor
for residence time and sediment flux. Elevational differences in
hydrogeomorphic disturbance regime and shorter residence time
at a more recently burned site at high elevation suggest that
changes in the frequency and severity of large storms and wildfire
could greatly alter floodplain sediment residence times and
associated downstream sedimentation.

## Results

**Radiocarbon ages and sediment residence times.** Calibrated
weighted mean radiocarbon ages of 52 charcoal fragment samples
ranged from 85 ± 66 to 6053 ± 70 cal y BP (Supplementary
Table 1). Individual weighted mean calibrated ages were mod-
erately correlated with sample depth ($r = 0.33$, $p = 0.023$). Site-
average mean calibrated radiocarbon ages of floodplain sediment
along four tributaries including Ouzel, Cony, Hunters, and North
Saint Vrain Creeks (Fig. 1) were moderately correlated, although
not significantly, with site-average charcoal sample depth at the
eight study sites ($r = 0.27$, $p = 0.55$; Table 1). The youngest mean
radiocarbon ages are not strictly from shallower depths (Sup-
plementary Fig. 1, 2; Supplementary Table 2), which suggests
reworking and spatiotemporal discontinuity of floodplain erosion

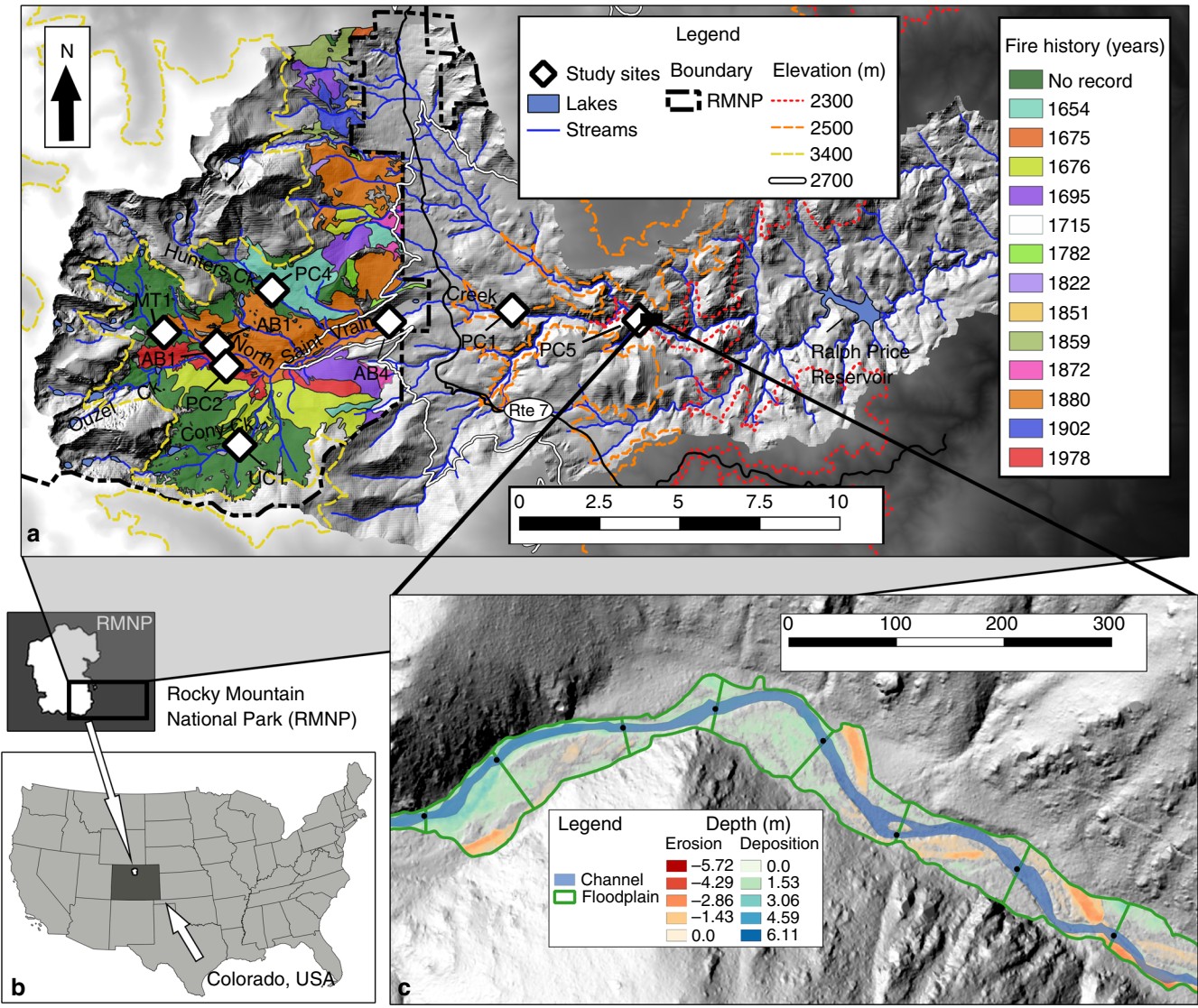

**Fig. 1** Map of the study area. Eight study sites located within the North Saint Vrain Creek watershed (**a**) in the southeastern portion of Rocky Mountain National Park (RMNP), Colorado, USA[68,69] (**b**), where fire history is well documented by Sibold et al. (2006)[32]. The scale bar from 1 to 10 denotes kilometers. Denoted elevations of interest include the previously proposed shift in floodplain disturbance regime (~ 2300 m)[14], the Pleistocene terminal moraine (~ 2500 m), the ecotone between montane and subalpine forests (2700 m), and the lower extent of the alpine zone (3400 m). An inset (**c**) illustrates an example of seven of the delineated 155 study reaches (denoted by bounding green lines) for which pre- and post-event lidar from State Route 7 to Ralph Price Reservoir was used to estimate floodplain erosion and deposition during the 2013 flood. The scale bar from 0 to 300 denotes meters

and deposition. We did not record evidence of a continuous sediment horizon indicative of a widespread fire or flood among the eleven transects and sampling locations at each study reach, but detailed stratigraphic observations necessary to identify such an event were not conducted. Charcoal sample locations lack the detailed stratigraphic context and apparent vertical continuity necessary to estimate floodplain sediment accumulation rates among samples (Supplementary Fig. 2). For these reasons, ages are not treated as indicators of specific fire or flow events. Instead, site-average ages are interpreted as a proxy for sediment residence time.

Age distributions at four out of five of the study sites had a sufficient number of radiocarbon ages ($n > 5$) to warrant examination (Fig. 2). Visual examination and comparison of coefficients of determination ($R^2$) and root mean square error (RMSE) values illustrate that exceedance probability is best explained by an exponential relationship with residence time in four out of the five sites examined (i.e., MT1, PC4, AB1, PC1).

This suggests that floodplain sediment is well mixed, has an equal probability of erosion within each study site, and that mean ages of several samples are a reasonable approximation of floodplain sediment residence time[29,38]. RMSE values support this observation for exponential relationships through values either smaller or very close to those for the power-law models (Fig. 2). Although the coefficient of determination ($R^2 = 0.94$) is lower and the RMSE (0.067) is higher for the exponential model of study site PC5, these values are very close to and also do well in describing the age distribution compared with the power-law model ($R^2 = 0.95$, RMSE = 0.054). Stronger performance of the power-law distribution model suggests that probabilities of evacuation of floodplain sediment at PC5 may be age dependent, such that younger deposits are more likely to be evacuated[29,38].

A series of more recent wildfires at a single study site (PC2; discussed in more detail below) appears to disrupt a strong longitudinal gradient in floodplain sediment residence time that otherwise results in older ages at higher elevations. Site-average

**Table 1 Field study site characteristics**

| Study reach | UC1 | MT1 | PC4 | PC2 | AB1 | AB4 | PC1 | PC5 |
|---|---|---|---|---|---|---|---|---|
| Pleistocene glacial history | Glaciated | Glaciated | Glaciated | Glaciated | Glaciated | Glaciated | Non-glaciated | Non-glaciated |
| Vegetation zone | Subalpine | Subalpine | Subalpine | Subalpine | Subalpine | Montane | Montane | Montane |
| Elevation (m) | 3054 | 3035 | 3013 | 2927 | 2901 | 2547 | 2420 | 2226 |
| Drainage area ($km^2$) | 12.7 | 14.8 | 10.1 | 11.1 | 19.1 | 82.2 | 105.3 | 243.8 |
| Valley width (m) | 26 | 61 | 15 | 14 | 67 | 247 | 33 | 27 |
| Valley length (m) | 55 | 100 | 40 | 60 | 90 | 215 | 130 | 170 |
| Channel width (m) | 6 | 7 | 4 | 5 | 7 | 14 | 17 | 13 |
| Confinement ($m\,m^{-1}$) | 0.22 | 0.11 | 0.25 | 0.36 | 0.10 | 0.06 | 0.51 | 0.46 |
| Channel slope ($m\,m^{-1}$) | 0.028 | 0.063 | 0.069 | 0.063 | 0.037 | 0.012 | 0.037 | 0.023 |
| Stream power ($W\,m^{-1}$) | 1740 | 3994 | 4183 | 3856 | 2439 | 1237 | 4319 | 4562 |
| Unit stream power ($W\,m^{-2}$) | 312 | 111 | 1140 | 753 | 357 | 88 | 252 | 358 |
| Sediment volume ($m^3$) | 402 | 1269 | 169 | 101 | 3126 | 28799 | 590 | 816 |
| Total reach mass (Mg) | 362 | 1142 | 152 | 91 | 2814 | 25919 | 531 | 734 |
| Mean sample depth (cm) | 35 | 28 | 38 | 27 | 36 | 41 | 39 | 24 |
| Charcoal Sample size, $n$ | 2 | 15 | 6 | 3 | 6 | 3 | 9 | 8 |
| Mean Age (cal y BP) | 1417 ± 880 | 1203 ± 545 | 1320 ± 512 | 178 ± 134 | 2108 ± 2140 | 1116 ± 551 | 668 ± 68 | 486 ± 443 |
| Flux per valley length (kg m$^{-1}$ yr$^{-1}$) | 16.2 | 8.3 | 6.0 | 2.8 | 7.1 | 98.7 | 4.0 | 7.9 |

Potential predictors for site averages of the weighted mean $^{14}$C ages and flux of fine floodplain sediment per valley length. Site-average ages are presented with standard deviations of the mean as errors

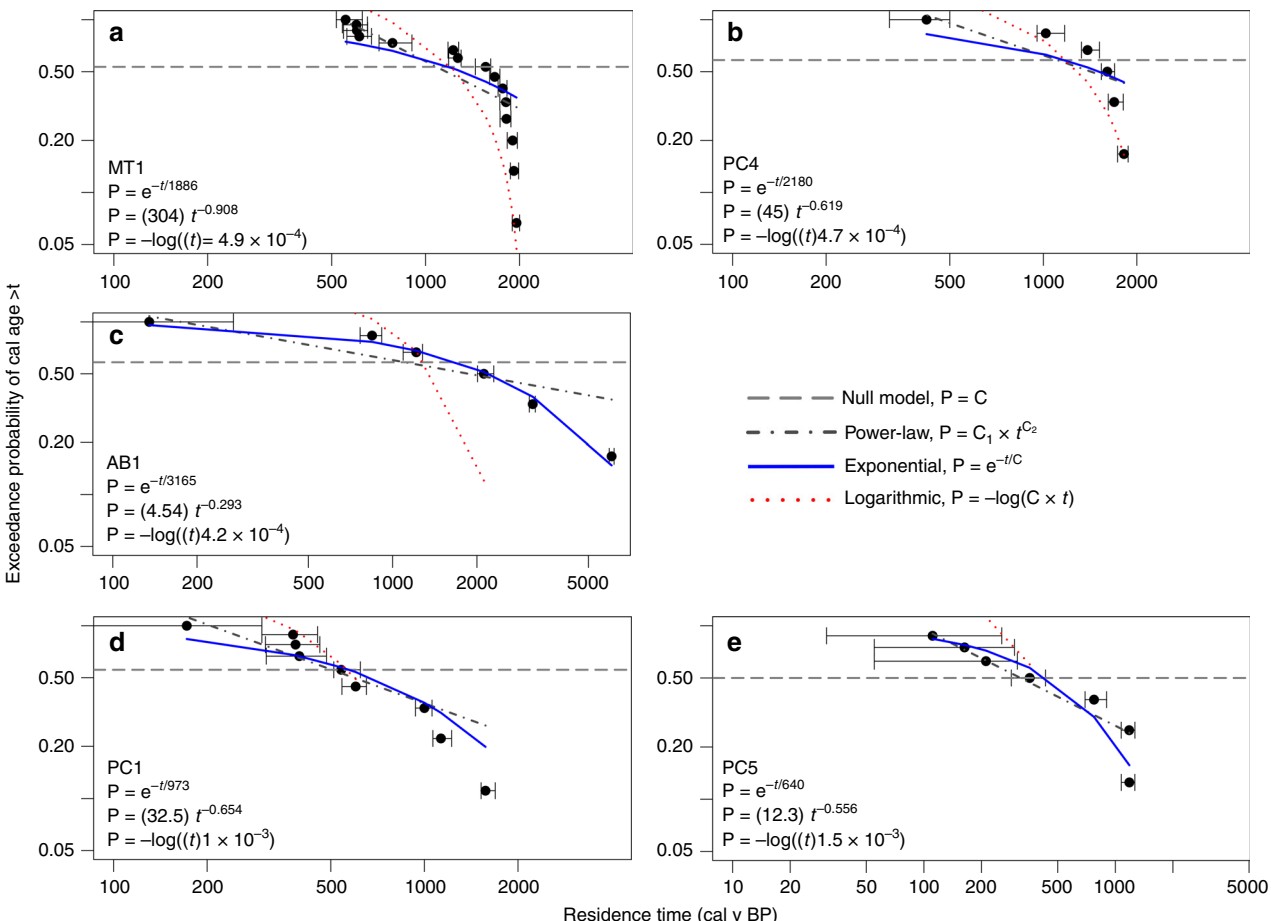

**Fig. 2** Exceedance probability (P) of weighted mean calibrated $^{14}$C ages. Plots are presented for sites with a sufficient number of ages to warrant examination of the age distributions as an estimate of residence time ($t$). Solid black circles indicate weighted mean ages and error bars represent the minimum and maximum values for each sample at the 95.4% confidence level. Equations for power-law, exponential, and logarithmic fits where $R^2$ represents the proportion of variance explained by a linear regression between predicted and observed values of P and RMSE is root mean square error. The exponential, power-law, and logarithmic models resulted in $R^2 = 0.89$, 0.82, and 0.88 and RMSE = 0.154, 1, 0.238 for MT1 **a**, $R^2 = 0.83$, 0.67, and 0.76 and RMSE = 0.174, 0.164, and 0.292 for PC4 **b**, $R^2 = 0.99$, 0.80, and 0.91 and RMSE = 0.037, 0.130, and 0.938 for AB1 **c**, $R^2 = 0.94$, 0.85, 0.94, and RMSE = 0.109, 0.113, 0.371 for PC1 **d**, and $R^2 = 0.94$, 0.95, and 0.96 and RMSE = 0.067, 0.054, and 0.655 for PC5 **e**, respectively. Source data are provided as a Source Data file

calibrated radiocarbon ages range from $178 \pm 134$ to $2108 \pm 2140$ years (Table 1). When the more recently burned PC2 site is excluded (25–240 calibrated y BP), a stronger positive linear

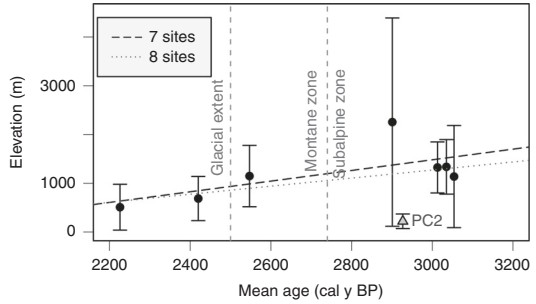

**Fig. 3** Site-average $^{14}$C ages with elevation. Error bars indicate the standard deviation for site-average calibrated mean ages, which is denoted by black circles for seven sites and a gray triangle for PC2. Variability explained ($R^2$) and $p$ values for the regression model of all eight study sites ($R^2 = 0.18$, $p = 0.29$) shows a stronger relationship for the seven study sites excluding site PC2 ($R^2 = 0.43$, $p = 0.11$)

correlation with elevation exists ($R^2 = 0.43$, $p = 0.11$; Fig. 3; Table 1). We examine and briefly discuss floodplain sediment dynamics across the longitudinal gradient, excluding the potential impacts of the single, more recently burned study site, before placing these results in the context of wildfire.

Residence time, stream power, and valley confinement vary significantly between six glaciated (of which five are in the subalpine and one in the montane zones) and two non-glaciated, montane zone sites (Fig. 4). The best multiple linear regression model (equation 1; adjusted $R^2 = 0.94$, $p = 0.037$)

$$t_a = 5.04 \times 10^{-5} - 1.05 \times 10^{-8}z - 4.64 \times 10^{-7}d + 2.16 \times 10^{-5}C_v - 8.39 \times 10^{-10}\Omega \quad (1)$$

indicates that elevation ($z$; $p = 0.067$), mean sample depth ($d$; $p = 0.077$), valley confinement ($C_v$; $p = 0.096$), and stream power ($\Omega$; $p = 0.43$) are the best predictors of average floodplain sediment residence time ($t_a$) at the seven study sites (Table 2). This indicates that relatively wider valleys at higher elevations appear to retain floodplain sediment at greater depths for longer durations than lower elevation sites. Channel slope ($S$; $p = 0.016$), valley confinement ($C_v$; $p = 0.019$), stream power ($\Omega$; $p = 0.033$), and mean sample depth ($d$; $p = 0.048$) are the best

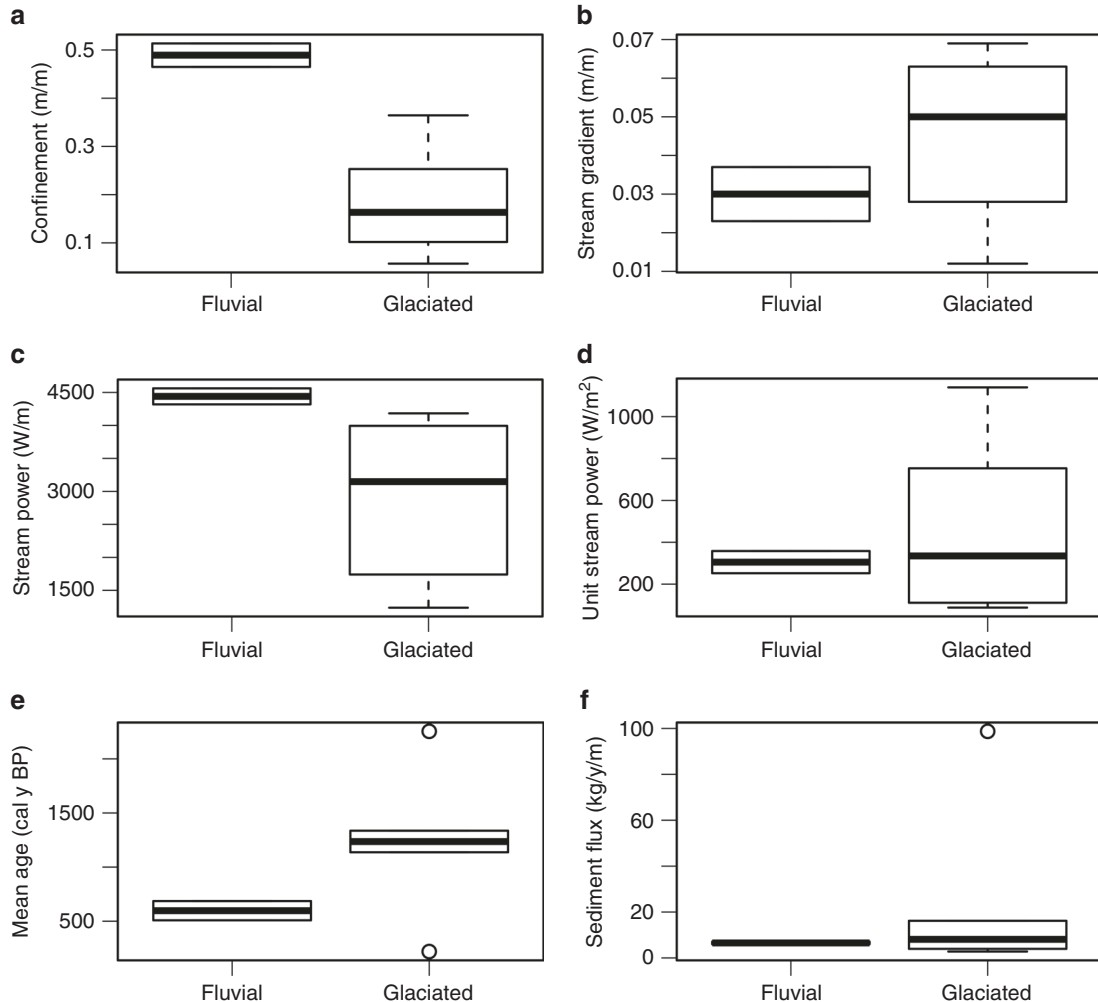

**Fig. 4** Boxplots of potential predictor variables. *P* values from Welch's *t* test for significant differences in means between glaciated ($n = 6$) and non-glaciated (fluvial; $n = 2$) study sites are $p = 0.001$ for confinement (**a**), $p = 0.251$ for stream gradient (**b**), $p = 0.031$ for stream power (**c**), $p = 0.412$ for unit stream power (**d**), $p = 0.063$ for mean age (**e**), and $p = 0.331$ for sediment flux (**f**). Thick lines represent the median (50th percentile), upper and lower edges of the boxes represent the 25th and 75th percentile. Whiskers extend to the minimum and maximum values within 1.5 times the interquartile range and open circles are outliers exceeding that range. Source data is listed in Table 1

**Table 2 Multiple linear regression models**

| Regression model details | Mean residence time ($t_a$) | Sediment flux ($f_s$) | Net transport ($T_n$) |
|---|---|---|---|
| Power transform[a] | − 1.717172 | − 0.020202 | 0.1576 |
| Intercept | 5.04E-05 | 1.01 | 1.30 |
| Elevation ($z$) | − 1.05E-08 | NA | NA |
| Confinement ($C_v$) | 2.16E-05 | 2.11E-01* | 1.44E-01**** |
| Channel slope ($S$) | NA | 1.77* | 6.12** |
| Stream power ($\Omega$) | − 8.39E-10 | − 2.90E-05* | NA |
| Mean depth ($d$) | − 4.64E-07 | − 2.22E-03* | NA |
| Upstream net transport ($T_{n-1}$) | NA | NA | 1.67E-04*** |
| Drainage area ($A$) | NA | NA | 2.82E-03*** |
| Shapiro–Wilk normality ($p$ value) | 0.850 | 0.468 | 0.123 |
| Non-constant variance test ($p$ value) | 0.448 | 0.564 | 0.613 |
| Adjusted $R^2$ | 0.943 | 0.941 | 0.445 |
| Regression model ($p$ value) | 0.037 | 0.039 | 2.20E-16 |

[a]"Values for the exponent used in the power transformations of the response variable. Variable coefficients, $p$ values for assumption tests, and model output statistics for the optimal multiple linear regression models for residence time, sediment flux, and net transport. Level of significance for each variable coefficient indicated by: ***0.0001, **0.001, *0.01, 0.05

predictors (equation 2) for the flux per valley length of floodplain sediment ($f_s$) through each reach (adjusted $R^2 = 0.94$, $p = 0.039$; Table 2).

$$f_s = 1.01 - 1.77S + 0.211C_v - 2.90 \times 10^{-5}\Omega - 2.22 \times 10^{-3}d \tag{2}$$

**Wildfire and sediment residence time.** Mapped wildfire ages from dendrochronology and fire-scarred trees by Sibold et al.[32] near the PC2 study site along Ouzel Creek provide important insight into the influence of wildfire on floodplain disturbance. A fire that occurred across the AB1 study site in 1880 falls within a period of increased probability of the summed charcoal age distributions in floodplain sediment at PC2 and AB1 and is mapped adjacent to the Ouzel Creek 1978 fire and PC2[32] (Fig. 5b, c). The charcoal records in sediment at PC2 and AB1 also have increased probability of charcoal ages spanning the period in which fires occurred in 1695 and 1715. Neither PC2 nor AB1 contain evidence with high probability of an older fire, which occurred in 1676 on the south side of the basin and adjacent to the boundaries for the 1978 fire (Fig. 5b, c). Although study site UC1 is located in old-growth forest without documented wildfires in the last 200 years, it is located close to the boundary of the 1676 fire and contains a small increase in probability of radiocarbon ages at that time. Otherwise, evidence for these more recent fires ( < 400 y BP) is not present in other surrounding high-elevation basins, which are located in old-growth forest (i.e., MT1, PC4, UC1). A possible signature of the 1715 fire, however, is preserved in the floodplain sediment at the two lowest-elevation study sites downstream from all other sites (PC1, PC5; Fig. 5c).

Radiocarbon ages presented here extend beyond the mapped fire history to provide evidence of localized fires in disparate basins as well as more widespread fires, evidence for which is preserved in floodplains at lower elevations (Fig. 5). Four high-elevation ( > 2600 m) study sites spread over three tributaries across the basin (MT1, UC1, AB1, and PC4) are mapped within old-growth forest that has not experienced a stand-replacing fire in over 400 years[32]. Two of these high-elevation sites, however, contain evidence of fires within the last 400 years (UC1 and AB1). The four highest elevation sites all contain evidence of older fires in floodplain sediment, and suggest much longer floodplain sediment residence times (mean ages > 1100 cal y BP) than lower elevation sites ( < 700 cal y BP). Increased probability of radiocarbon ages between 1800 and 1900 cal yr BP across higher

elevation study sites (MT1, UC1, PC4) and AB4 suggests a widespread large fire or numerous small fires during that period, which coincide with drought conditions in North Dakota from ~ 1 to 400 CE and New Mexico from ~ 100 to 150 CE, as summarized by Woodhouse and Overpeck[40].

The presence of older charcoal and a wider spread in ages at higher elevation sites indicate increased potential for long-term storage, whereas limited old charcoal and decreased variability centered around young mean ages at lower elevations suggest more rapid turnover and decreased residence time of floodplain sediment. Variability in ages at high-elevation sites ( > 2500 m) is higher compared with lower elevation sites, and summed probability distributions are not long-tailed or skewed; probabilities are relatively consistent across the peaks in the summed probability distributions (Fig. 5). Study sites below 2500-m elevation (i.e., PC1, PC5), however, exhibit long tails with the highest probabilities at very young ages. Although the number of charcoal fragment samples for each site were limited in this analysis ($n = 6$ to 15), exceedance probabilities for residence times support these observations by indicating that higher elevation sites with a sufficient number of samples exhibit an exponential relationship (Fig. 2)[29,38,39]. The similarity in the performance of the exponential and power-law distribution models, however, suggest that more charcoal fragment samples would provide a more robust analysis of the residence time distributions.

Although mean ages of high-elevation old-growth sites and the more recently burned PC2 site differ significantly at the 90% confidence level ($p = 0.063$; Fig. 4), the mean for PC2 is based on only three samples. Although this small sample size could fail to capture older charcoal present in the floodplain, the other four high-elevation sites contain only limited records of fires that occurred < 400 cal y BP (Fig. 5).

**Floodplain deposition and observations from the 2013 Flood.** Of the net $106,600 \pm 37,600$ m$^3$ of sediment eroded and/or deposited along the floodplain of 155 study reaches between 2515 and 1950-m elevation during the 2013 flood, > 76% occurred below 2215-m elevation (Fig. 6). We present net change in surface elevation and do not account for deposited sediment that was later eroded between the two lidar flights or where sediment was deposited following the erosion of sediment throughout the duration of the event. Net change along the floodplain following the 2013 flood was dominated by $146,000 \pm 52,000$ m$^3$ of deposition, although much of the deposited sediment was later

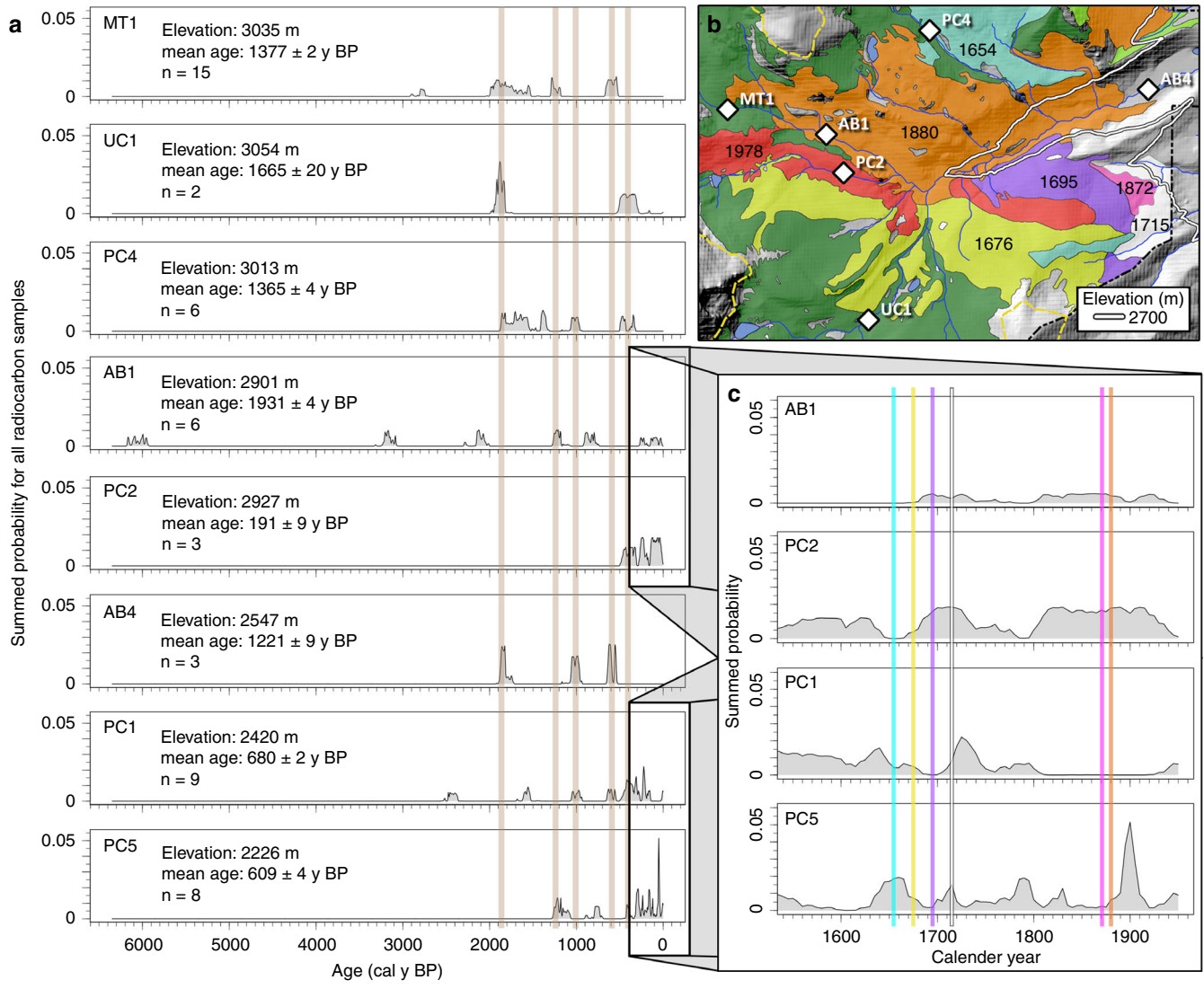

**Fig. 5** Normalized summed probabilities of $^{14}$C ages. Summed probabilities of all large charcoal radiocarbon samples from each study reach (**a**). Tan shaded areas indicate periods of interest where fires appear to be recorded in more than one study site. The inset (**c**) displays summed probabilities of radiocarbon ages after 1500 CE compared with tree-ring dated wildfire ages (**b**). Vertical lines in (**c**) are color-coded to match wildfires mapped in Figs. 1b and 5b[32]. The white contour line (**b**) represents 2700 m elevation, which is the upper extent of the montane zone. White diamonds represent the location of field study sites (**b**), and study sites PC1 and PC5 are located downstream from all sites in this inset. Source data are provided as a Source Data file

remobilized during the 2014 annual peak flow[41]. The net volume of sediment eroded from the floodplain ($39,000 \pm 14,000$ m$^3$), was < 30% of the deposition that occurred. Notable increases in floodplain disturbance occurred at the confluences of N. St. Vrain Creek and tributaries. In many cases, additional sediment input to the floodplain coincided with debris flows and landslides on adjacent slopes and within tributaries. The proportion of cumulative floodplain disturbance remained constant for short distances along N. St. Vrain Creek where the valley was highly confined. The relative volume of floodplain sediment disturbed increased where valleys are less confined (Fig. 6).

The difference between the amount of sediment delivered to each reach (from hillslopes and upstream reaches) and the amount transported from that reach (net transport, $T_n$) is most highly correlated with the net transport of the upstream consecutive reach. Multiple regression indicates that $T_n$ is best explained by drainage area ($A$, $p < 0.001$), channel slope ($S$, $p = 0.004$), valley confinement ($C_v$, $p < 0.001$), and the net transport from the consecutive upstream reach ($T_{n-1}$) (equation 3; $R^2 =$

$0.45$, $p < 0.001$).

$$T_n = 1.3 + 1.67 \times 10^{-4} T_{n-1} + 3.82 \times 10^{-7} A + 0.144 C_v + 6.122 S$$

(3)

These results highlight the importance of valley confinement both at a site of interest and along adjacent upstream river segments in regulating the geomorphic response to large floods in the study area (Table 2).

## Discussion

Results support the hypothesis that floodplain sediment residence time is significantly longer along higher elevation, glaciated valleys in the CFR compared to low-elevation non-glaciated sites (90% confidence level; Fig. 4). However, elevation alone does not explain the observed variability in the study area. Localized valley geometry and geomorphology are required to explain significant variability in differences between sediment residence times and fluxes. Valley confinement, stream power, and average sample

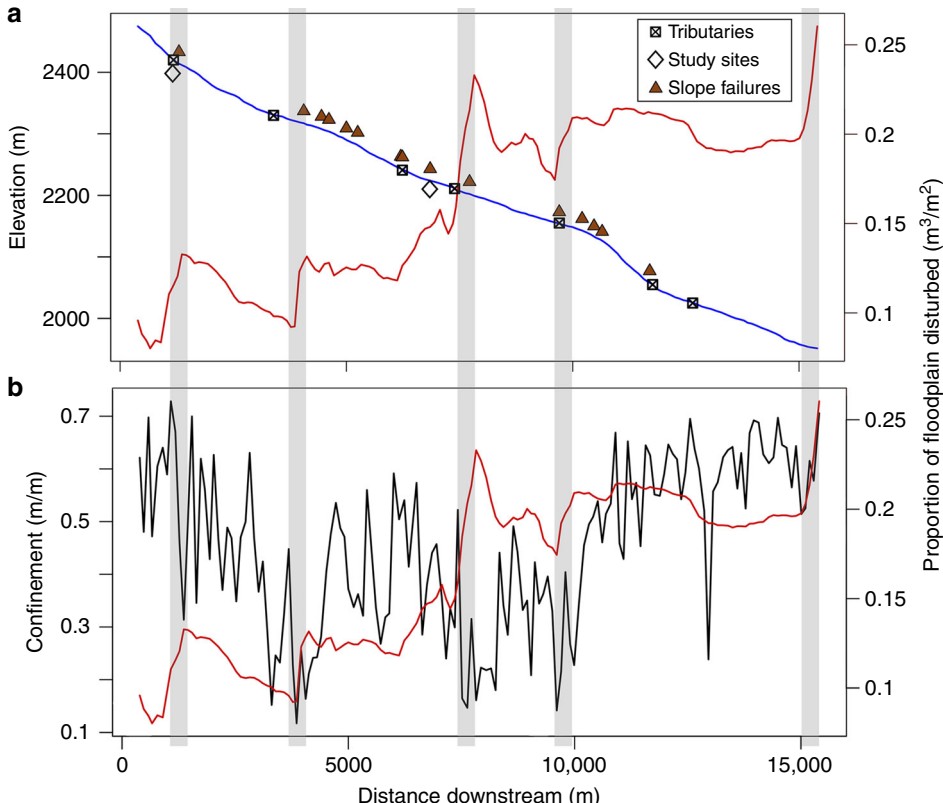

**Fig. 6** Observations of 2013 flood from lidar analysis. Proportion of disturbed floodplain (cumulative volume displaced divided by cumulative floodplain area) with increasing distance downstream along North Saint Vrain Creek is indicated by the red lines (**a**, **b**). The location of tributary junctions, field study sites, and slope failures are indicated along the blue line, which indicates decreasing elevation (**a**). Changes in valley confinement measured at 155 study reaches along North Saint Vrain Creek are indicated by the black line (**b**). Shaded areas indicate regions of increasing floodplain disturbance, commonly coinciding with decreases in valley confinement, tributary junctions, and slope failures, at which deposition of sediment was the dominant response. Source data are provided as a Source Data file

depth are predictors of both floodplain sediment residence times and sediment fluxes at the study sites (Table 2). Apparent zonation of floodplain disturbance may reflect hydrogeomorphic response between the subalpine and montane zones, degree of valley confinement associated with glaciation, and the upper elevation limits of convective storms in the CFR[13,42] (Fig. 7).

Our estimates of floodplain sediment residence time suggest that floods of the observed magnitude in 2013 occur on the order of $10^2$ to $10^3$ years at lower elevations, which may not occur at the same frequency and magnitude or simply do not cause the same magnitude of hydrogeomorphic response at higher elevations. Valley confinement varies significantly at the 90% confidence level between glaciated and non-glaciated study sites (Fig. 4) and is a significant predictor for floodplain sediment residence time and flux (Table 2). Confinement and stream power are significantly correlated ($R^2 = 0.69$, $p = 0.09$; Supplementary Table 3, 4) at the study sites and confinement is a significant predictor ($p < 0.001$) of the net volume of sediment eroded from consecutive reaches during the 2013 flood. These results suggest that differences in floodplain sediment residence time and flux between glaciated and non-glaciated sites likely reflect increased ability of wide valleys to dissipate energy, accumulate sediment, and leave floodplains intact during large floods. Although relatively unconfined sites at high and low elevations dissipate energy and result in the accumulation of sediment, the influence of cumulative discharge at lower elevations sites likely results in the disturbance of preexisting floodplain sediment. Other transitions along this boundary, however, are likely to significantly influence floodplain disturbance and sediment dynamics, most notably

hillslope connectivity and wildfire. Before discussing these factors, it is important to address potential error and the assumptions of our approach.

Use of charcoal radiocarbon ages as a proxy for the timing of sediment deposition can be problematic because of inbuilt ages as a result of "old wood"[31] and potential reworking of charcoal [28]. "Old wood" refers to the fact that using charcoal to date the age of a fire can result in erroneous inbuilt ages from inner wood of trees or charred dead wood because the tree may have already died prior to a fire at a much later date[31]. The potential magnitude of error from inbuilt ages of old wood are dependent upon decomposition rates of specific species in a given climate, which regulate the turnover time of dead wood. Turnover time of softwood tree species in our study area are on the order of ~ 150–900 years[43], meaning that radiocarbon ages may overestimate the timing of a fire by up to 900 of years. The potential for inbuilt ages that could produce errors in radiocarbon ages on the order of 500 years[31], creates less separation between high and low-elevation sites. However, differences in residence times between low and high-elevation sites (~ 1000 y) fall outside of this range of error. In addition, the highest standard deviations are present at high-elevation sites, whereas low-elevation sites have much less variability in ages. This decrease in variability at low elevations can be interpreted to provide support for shorter residence times at lower elevations. Because we do not use charcoal to reconstruct fire history or to identify distinct sedimentation events, the largest component of error associated with inbuilt ages simply compounds those associated with reworking of charcoal.

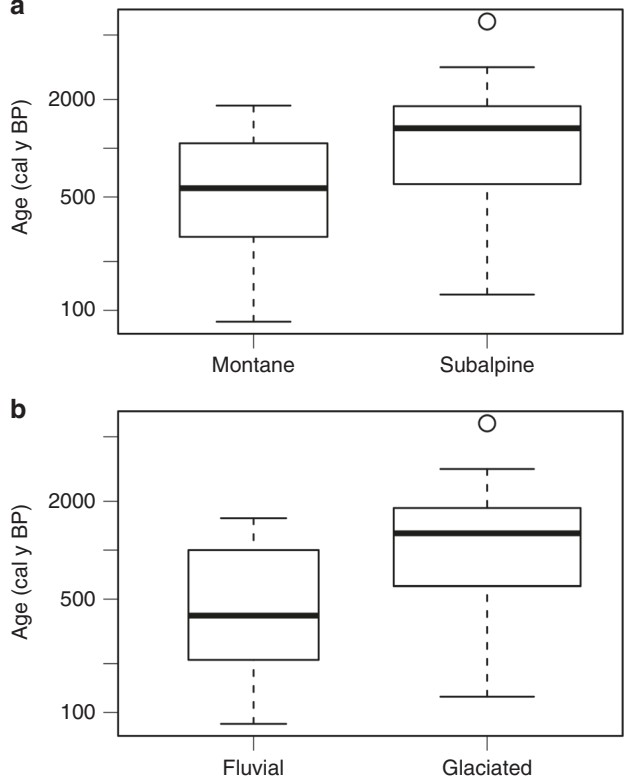

**Fig. 7** Boxplots for mean age from adjusted *t* tests. All 52 calibrated ages between all study sites are included to examine differences in the montane and subalpine zones (**a**) and those upstream (glaciated) and downstream (fluvial) of the Pleistocene terminal moraine (**b**). *P* values are 0.003 and 0.0007, respectively. Thick lines represent the median (50th percentile), upper, and lower edges of the boxes represent the 25th and 75th percentile. Whiskers extend to the minimum and maximum values within 1.5 times the interquartile range and open circles are outliers exceeding that range. Source data are provided as a Source Data file

Reworking of charcoal on hillslopes and along river corridors through episodic erosion and deposition limits interpretation of charcoal radiocarbon ages as a proxy for the timing of sediment deposition. Error associated with reworking exists as a lag between the death of a tree and the date charcoal was deposited in floodplain sediment[28]. That lag associated with reworking is likely to be shorter in our relatively small CFR study basin ( < 200 km²) with more rapid turnover time of softwood compared to that of charcoal from hard wood (*Angophora* and *Eucalyptus*) in a ~ 2000 km² basin by Blong and Gillespie[28].

Increases in error from potential lag associated with both inbuilt ages and reworking decrease the support for the hypothesis that residence times are longer at higher elevations. Errors are compounded with increasing basin size, such that downstream study sites with larger drainage areas (i.e.,10² km² compared with 10 km²) are likely to have charcoal with older ages as a result of reworking and sporadic transport. Thus, increased error introduced by inbuilt ages and reworking of charcoal samples in our study area would not contribute to a false support of our hypothesis (a type 1 error), but would make it more difficult to observe a significant difference to support our hypothesis. However, special care was taken to date relatively large ( > 1 cm³), loose charcoal samples, so breakdown of larger softwood charcoal pieces during transport decrease the likelihood that older charcoal from greater distances upstream would be sampled in floodplains at the lowest-elevation sites. For these reasons, a more specific

interpretation of our ages is perhaps residence times within the contributing watershed, which we would expect to increase with increasing drainage area.

In situ production of charcoal on the floodplain from localized fires is another source of error that may result in artificially young ages. Combined with other ages, which represent a maximum age given potential error for reworked charcoal, in situ charcoal production can help develop constraints on the timing of deposition. These minimum age constraints, however, require observation of a continuous charcoal layer in floodplain stratigraphy as evidence for fire occurrence. Although we did not identify evidence of a system-wide or even localized wildfire across individual study sites, our surveys lacked the stratigraphic resolution to examine appropriate stratigraphy in detail.

Using site-average weighted mean sediment ages as a proxy for residence time requires the assumption of steady state conditions, which we relax in our analysis and interpretation. Because floodplain sediment flux and residence times are a function of spatiotemporal variability in erosion and deposition associated with changes in climate, hydrologic regime, land cover, wildfire, extreme events, and slope failures, this assumption cannot be fully met.

When older sediment is less likely to be eroded, mean age is likely an overestimate of sediment residence time[39,44], but in a well-mixed reservoir mean age is approximately equal to the residence time[38]. In the relatively confined 1st and 2nd order streams examined in this study, floodplain sediment residence times are likely on the order of $10^2$–$10^3$ years. This is much shorter than residence time modeled along a meandering river where long-tailed distributions of much older sediment ($10^3$–$10^4$ y) have a decreased probability of erosion along large rivers[44]. Systematic random sampling employed in this study is assumed to capture the variability in estimates of site-average ages given continuous erosion and deposition along floodplains of each study site. In addition, our analysis of probability distributions presented in the results to infer relative mixing of floodplain sediment supports the assumption of well-mixed reservoirs and site-average ages as a proxy for floodplain sediment residence time. Investigation into the causes of differences in residence times requires details about forest type, hydrogeomorphic response, and wildfire regime.

Different forests types in the study area exhibit differences in hydrologic runoff response, which can be linked to floodplain disturbance and residence times. The montane zone receives an annual mean precipitation of ~ 55 cm[45], is dominated by Douglas fir (*Pseudotsuga menziesii*) and ponderosa pine (*Pinus ponderosa var. scopulorum*)[46], and experiences low severity fires with a recurrence interval of 30–100 years[32,46,47]. In the montane zone, increased potential for intense convective storms, decreased ground cover and forest density on south-facing slopes of ponderosa pine dominated forests, more highly confined river valleys[46,48], more rapid runoff response to precipitation, and a cumulative effect of tributaries joining along incised canyons, have resulted in extreme floods historically observed and evidenced by paleoflood indicators[36,42]. The subalpine zone receives an annual average precipitation of ~ 77 cm[45], is characterized by subalpine fir (*Abies lasiocarpa*), limber pine (*Pinus flexilis*), lodgepole pine (*Pinus contorta*), Engelmann spruce (*Picea engelmannii*), and aspen (*Populus tremuloides*), and experiences less frequent, large, stand-replacing fires on average every 500 years[32,46,47]. In the subalpine zone, denser forest canopies and ground cover increase interception and slow runoff response to rainfall, creating a buffer between typical afternoon thunderstorms and resulting potential for large floods. The alpine zone receives an annual average precipitation of ~ 102 cm, which typically falls as snow in the winter and isolated convective

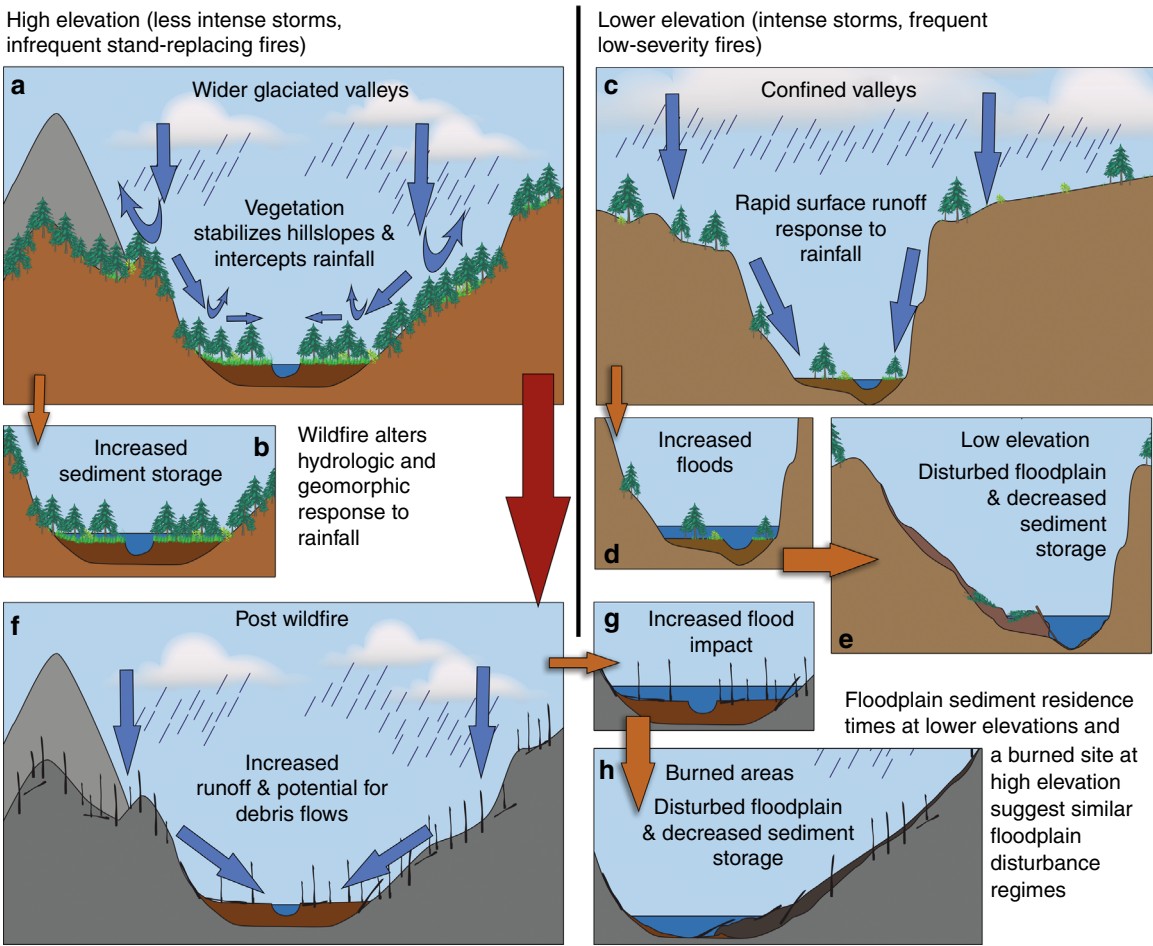

**Fig. 8** Forest characteristics and hydrogeomorphic response. Conceptual diagram illustrating the influence of differences in hydrologic and geomorphic response on floodplain sediment residence time along partially confined and unconfined valleys in the Colorado Front Range. Abundant vegetation in the subalpine zone with low fire frequency intercepts rainfall and slows runoff response, which stabilizes hillslopes (**a**) and results in stable floodplains and sediment storage (**b**). Tighter coupling between hillslopes, less vegetation cover, increased fire frequency, increased likelihood of large convective thunderstorms, and more rapid rainfall–runoff response in narrowly confined valleys in the montane zone (**c**) result in increased flood magnitude (**d**), increased potential for debris flows and slope failures, and more frequent floodplain disturbance (**e**). Decreased vegetation cover and interception and increased rainfall–runoff response in the subalpine zone following wildfire (**f**) results in increased flood magnitude (**g**), potential for debris flows, and floodplain disturbance, and decreases in resulting floodplain sediment residence time (**h**). Images depict differences in geometry and hydrogeomorphic response along relatively partially confined and unconfined valley segments representative of study sites at high and low elevation. However, charcoal samples were not analyzed from highly confined sites with little to no floodplain present. Although more confined valley segments exist at both high and low elevation, valleys below the Pleistocene terminal moraine typically do not contain wide unconfined valleys in the study area

thunderstorms in the summer months[45]. In the alpine zone, vegetation is characterized by alpine tundra, grasses and willow (*salix spp.*), and Krummholz subalpine fir (*Abies lasiocarpa*) and Krummholz Engelmann spruce (*Picea englemannii*). With limited fuel and lack of trees to reconstruct fire history, there is insufficient data about fire regimes in the alpine zone.

Riparian vegetation community structure in the CFR typically reflects the surrounding forest zones in which the community is located, but can differ in relatively unconfined valleys and where beavers are present. River corridors with widths > 20 channel widths or with evidence of beaver activity including ponds, dams, beaver canals, berms, or buried dams (i.e., AB1, AB4), exhibit distinct differences in vegetation and are dominated by grasses and sedges (*Carex* spp.), willow (*Salix* spp.), and river birch (*Betula fontinalis*)[46,49]. Riparian vegetation in the subalpine zone can include aspen (*Populus tremuloides*), Douglas fir, Engelmann spruce, and blue spruce (*Picea pungens*)[49].

Elevational differences of residence times in the study area likely reflect differences in forest type and associated hydrologic runoff response and geomorphic disturbance, which are greatly altered by wildfire. Because wildfire greatly changes hillslope–floodplain connectivity and hydrogeomorphic response to rainfall, which our results suggest may influence floodplain stability and sediment residence times. Loss of forest floor cover, thinner forest canopies, and reduced infiltration of rainfall following wildfire result in rapid hydrologic response[50], increased flood magnitudes[51], increased potential for debris flows[52], abundant erosion[53], increased likelihood of slope failures[52,54], and increased potential for hillslope sediment to reach valley bottoms in burned areas[21] of the subalpine zone (Fig. 8). Similarly, less vegetation cover and resulting rapid response to rainfall, increased cumulative flood magnitude[14,33], increased hillslope connectivity[10], and likelihood of slope failures at lower elevations in the CFR[35] are reflected in differences of lower floodplain sediment residence times at low-elevation sites compared with those at higher elevation. Slope failures at lower elevations have potential to dominate denudation of the CFR and may be a significant factor in floodplain erosion in the montane zone[17].

Results presented here quantify differences in residence time of floodplain sediment in the montane zone and higher elevations and as well as spatial heterogeneity of valley geometry and the ability of rivers to transport hillslope-derived sediment.

Estimates made from the DoD analysis presented here incorporated all valley bottom sediment, which included gravel and boulders displaced during the flood, whereas estimates made from field surveys and radiocarbon analysis involved only fine floodplain sediment above cobbles and boulders. Thus, the lidar analysis includes underlying gravel, which would over-estimate fine floodplain sediment displacement, but our results provide meaningful insight into potential factors that influence floodplain disturbance and sedimentation.

Proposed elevation thresholds and longitudinal trends in slope failures and floodplain disturbance of the CFR vary[35,36,55], but are likely indicative of a shifting threshold or a long-term average relating to several factors mentioned above, including precipitation, forest type, and runoff response to rainfall. This is supported by differences in the proposed long-term floodplain disturbance elevation thresholds at 2300 m[14] and the inferred debris flow and landslide occurrence threshold at ~ 2600 m during the 2013 event[35]. Mean radiocarbon age and inferred floodplain sediment residence times presented here from the N. St. Vrain Creek watershed suggest a transition in the frequency of floodplain disturbance between wider glaciated and partly confined ($C_v <$ 0.55) unglaciated river corridors above and below 2500 m in elevation, respectively. Considering these elevation ranges in the context of the hydroclimatic shift in flood response, we suggest that any such boundary likely varies across basins and falls within a range that captures variability in extreme events.

Changes in the magnitude and frequency of extreme events may cause the elevational hydrogeomorphic disturbance boundary to shift upward under a changing climate in the CFR, similar to the observed up-slope shift in the snow–rain transition[18]. This means that increased frequency and magnitude of both wildfire and extreme events anticipated to accompany climate change are likely to greatly alter sediment dynamics, floodplain stability, and the fate and transport of carbon and contaminants in mountain watersheds of the Colorado Front Range.

Our results indicate that floodplain sediment residence times are longer within high-elevation glaciated river corridors compared with unglaciated mountain streams at lower elevations of the Colorado Front Range. Together with observations from the 2013 Front Range floods, radiocarbon ages strongly suggest that valley confinement is a significant control on the heterogeneity of floodplain disturbance and likely influences observed elevation differences in residence times. Less-confined glaciated valleys are conducive to floodplain storage, which result here in longer floodplain sediment residence times, showing that glaciated mountain valleys can serve as long-term sediment reservoirs. Observed shifts in boundaries between precipitation regimes[18] and an increase in the severity and frequency of extreme floods[11] and wildfire[22] are likely to greatly alter sediment dynamics, floodplain stability, and sediment residence times in the Colorado Front Range. Because long-term trends in precipitation patterns, extreme floods, and wildfire regimes influence floodplain sediment regimes, climate change could greatly alter sediment dynamics in river corridors. Previously-glaciated, high-elevation, mountainous floodplains that are relatively stable, retentive sediment reservoirs at time spans of $10^2$–$10^3$ years under existing climate could become net sediment sources to rivers and downstream reservoirs. This can have important implications for ecosystems and water quality with respect to sediment yield, nutrient cycling, carbon dynamics, and the fate and transport of contaminants stored along river corridors at high elevations.

## Methods

**Study area**. Like many mountainous regions globally, the CFR, USA is characterized by a shift in precipitation and hydroclimatic disturbance regime with increasing elevation[13,56]. The entire CFR experiences an annual snowmelt-driven peak flow, but elevations below ~ 2300 m sometimes experience secondary peak flows from convective rainfall[14]. Extreme storm events in the CFR sometimes occur in the summer and fall as a result of increased moisture transport from the Gulf of Mexico and convective storms that are stalled over the mountains through a variety of atmospheric mechanisms[13,42].

Study sites also span the proposed shift in hydroclimatic regime (2300 m) and the Pleistocene terminal moraine (~ 2515 m)[46]. Because hillslope hydrology and runoff processes are influenced by vegetation type, precipitation patterns, and topography, all of these boundaries have potential to influence differences in hydrogeomorphic floodplain disturbance across this elevational gradient. The hydroclimatic elevation threshold at 2300 m proposed by Jarrett[36] may vary spatially across basins and may shift over time with changes in climate and hydrologic regime. However, no distinct changes in hydrologic response of N. St. Vrain Creek occur with downstream hydrologic geometry. Past work shows that annual peak flow ($Q$ in ft$^3$ s$^{-1}$) along N. St. Vrain Creek scales linearly in the basin with drainage area ($A_d$; in mi$^2$) by Equation (4)[57].

$$Q = 5.5A_d + 197 \qquad (4)$$

**Study sites**. Vegetation at all study sites reflected vegetation typical of riparian areas in their respective vegetation zone, such that subalpine fir, limber pine, and primarily blue spruce were common in the subalpine zone and ponderosa pine was common in the montane zone. Study site AB4, in the montane zone and located immediately upstream of the Pleistocene terminal moraine, is a > 200 m-wide beaver meadow complex, and was dominated by sedges, grasses, willow, and river birch. Surrounding uplands at AB4-contained vegetation characteristic of a transition between the montane and subalpine zones, with a mixture of ponderosa pine, lodgepole pine, Engelmann spruce, aspen, and Douglas fir. A narrower meadow study site in the subalpine zone, AB1, also exhibited increased abundance of grasses and willow, but contained an abundance of blue spruce.

All sites were selected as representative river reaches with continuous valley characteristics for 10–20 times the channel width. Confinement was a primary characteristic considered such that all reaches were partly confined to unconfined river segments, with channel to valley width ratios below 0.55. Confined valley segments where channel to valley width ratios exceeded 0.55 were not included in this study because they lacked floodplains, sediment available for sampling, and charcoal samples. Additional sites were surveyed but did not contain sufficient charcoal samples > 1 cm$^3$ to be included in this study. Erosion and complete restructuring of the floodplain at elevations below 2500 m elevation during the 2013 flood limited the number of sites below that elevation. For these reasons, study sites in the montane zone and below the terminal moraine at 2500 m in elevation were limited in number.

Topographic surveys and measured depths of fine overbank sediments were used to calculate floodplain fine sediment volumes and the mass of sediment along each study site. Surface topography was surveyed along each study site at breaks in slope along each of the eleven transects with a maximum spacing less than 11 meters, as outlined in Sutfin and Wohl (2017)[4]. At every point where surface topography was surveyed, the depth of fine floodplain sediment was measured by inserting a 1.27-cm diameter and 1.8-m long soil probe into the floodplain surface until refusal at bedrock or cobbles. Sediment volumes were estimated using triangular irregular networks in ESRI ArcMap software. The total mass of fine floodplain sediment was estimated by dividing the mean measured bulk density (described below) by the volume of estimated floodplain sediment. The mass of sediment was divided by the residence time[37] and the length of the valley along each study reach for a first-order approximation of floodplain sediment flux per unit valley length.

Additional details regarding the methods for overbank sediment sampling along transects, topographic surveys, the statistical rationale for the number of sampling locations, and sites of more intensive sampling are provided in the main text and supplemental material in Sutfin and Wohl (2017)[4].

Floodplain sediment was identified primarily as fluvially transported overbank deposits because they were dominated by sub-rounded and sub-angular sand (~ 75%)[4] and gravel sorted into lenses containing various proportions of sand, gravel, silt, and organic matter. Evidence of past mass wasting events and observed debris flows, such as those that occurred in the 2013 floods, however, suggest potential reworking and deposition of hillslope sediment along valley bottoms, primarily in more confined reaches and those below the terminal moraine. Fluvial deposits were sought out and distinguished apart from areas that exhibited fine-grain matrix-supported, unstratified, and poorly sorted features that were likely debris-flow deposits from adjacent hillslopes. Although areas with debris-flow deposits were identified separately from the floodplain and floodplain study sites were selected to minimize the influence of debris flows and additional hillslope inputs, these mass movements could certainly still influence sediment examined along river corridors in this study. This is true particularly as debris flows appear to greatly influence floodplain stability at lower elevations in the study area[17,26,58].

**Sediment samples and radiocarbon analysis**. A random number generator was used to select at least one sampling location along each of the eleven transect at a random distance from the valley edge for all eight study sites. Where the random selection was located in the active river channel, on a boulder, or on bedrock another random location was selected from one side of a river channel segment without a sample. Fine floodplain sediment samples were collected using a stainless-steel hand auger for the full depth of the floodplain sediment profile at 15-cm increments. Two study sites, PC1 and MT1, were sampled more intensively resulting in 14 and 31 sampling locations, which were used to assess the variability of sediment characteristics for a separate study. Small pits ( < 50 cm) were dug to extract 33 horizontal bulk density samples using a 7.3-cm diameter and 7-cm long soil sampling tube at several floodplain locations in the study area to estimate an average bulk density of $0.91 \pm 0.22$ g cm$^{-3}$.

Each floodplain fine sediment sample was screened for charcoal fragments for radiocarbon analysis. All charcoal pieces were removed from each sample and a total of 52 large singular charcoal fragments ( > 1 cm$^3$) were dissected and cleaned under a low power microscope to remove detrital matter and fine rootlets. Cleaned samples were weighed and sent to a commercial lab, Direct AMS, for processing using accelerator mass spectrometry of $^{14}$C. OxCal software[59] and the IntCal13 calibration curve[60] were used to calculated weighted mean calibrated radiocarbon ages[61]. We use site averages of weighted means from between 2 and 15 calibrated radiocarbon ages obtained at each site as a proxy for floodplain sediment residence time. We refer to this as a site-average mean age, or simply as a mean age, which differs from individual weighted mean ages. This approach makes many assumptions, which we discuss including a closed system and steady state conditions[38] that we examine using residence time age distributions[29,38]. Results of this analysis can be used to infer relative mixing of floodplain sediment, and preferential probabilities of younger versus older sediment to be eroded from the floodplain[29,38]. Probabilities for all radiocarbon samples at each study site were summed and normalized to produce summed probability distributions for each study site.

Probability distributions were examined to infer relative mixing of floodplain reservoirs and assess the validity of using mean age as an estimate for residence time[38,39]. The distributions of floodplain sediment residence times were examined at each study site by normalizing the cumulative probability of the weighted mean ages (summed from older to younger) at the 95.4% confidence level. These normalized probabilities, referred to as exceedance probabilities, represent the likelihood that the study site has a residence time younger than the mean age of each respective radiocarbon age. After plotting the exceedance probabilities against the weighted means, we examined the shape of the probability distribution of the five study reaches with more than five radiocarbon samples to infer relative mixing of the floodplain reservoirs (Fig. 2)[29,39]. Best-fit lines were estimated using the *nlsLM* function in R to identify initial parameter values and run again to select final coefficients for power, exponential, and logarithmic model fits. Best-fit models were selected using the highest coefficient of determinant (R$^2$) values from a linear regression between the observed and predicted exceedance probabilities. Root mean square error (RMSE) was also calculated using the *RMSE* function in R and examined in comparison with R$^2$ values.

To examine potential controls on floodplain residence time and fine sediment flux, we calculated numerous variables along each of the eight field study reaches. Mean floodplain ($w_v$) and channel width ($w_c$), valley confinement ($w_c$ $w_v^{-1}$), stream gradient ($S$), stream power ($\Omega = \rho g S Q$), unit stream power ($\omega = \Omega$ $w_c^{-1}$), and mean sample depth ($d$) were calculated along each reach (where $\rho$ is the density of water and $g$ is the acceleration of gravity) for statistical analysis and for comparison with results from lidar differencing.

**Lidar differencing and analysis**. A digital elevation model (DEM) of difference (DoD)[62] with 0.75-m resolution and a geomorphic change detection error threshold of ±0.34 m was used to calculate erosion and deposition between 2012 pre-flood and 2013 post-flood lidar below 2515-m elevation, as previously described by Rathburn et al.[41]. DEM with 0.75-m resolution along the lower portion of the study basin were used from pre-flood lidar acquired through Boulder County in the state of Colorado and post-flood lidar acquired through the Federal Emergency Management Act. The maximum vertical error in the DoD was propagated through estimated sediment volumes to provide a maximum range of error in results presented.

The pre-event lidar and 2012 US Geological Survey (USGS) 1-ft resolution orthoimagery were used to delineate channel width based on the presence of water and unvegetated gravel bars. Flowlines generated from the USGS 10-m resolution DEMs were used to calculate cumulative drainage area and 1-m pre-event lidar was used for all other analyses. The *Fluvial Corridors* toolbox[63] was used to delineate the floodplain, which was segmented into 155 reaches ~ 100 m in length. Floodplain segments were used to extract floodplain and channel length and area, drainage area ($A$), and elevation. These parameters were used to calculate mean floodplain ($w_v$) and channel width ($w_c$), valley confinement ($w_c$ $w_v^{-1}$), stream gradient ($S$), stream power ($\Omega = \rho g S Q$), unit stream power ($\omega = \Omega$ $w_c^{-1}$), and the downstream gradient in stream power, where $\rho$ is the density of water and $g$ is the acceleration of gravity.

**Statistical analysis**. Statistical analysis was conducted in R statistical software[64]. All univariate correlation coefficients ($r$) listed here are Spearman correlations. Two-sided $t$-tests to test differences among high-elevation and low-elevation study sites were conducted using the $t$ test function in R for unequal variances, which runs a modified analysis for a Welch's $t$ test for unequal sample sizes and unequal variance.

Analysis included examination of the influence of valley and basin characteristics, channel geometry, and hydraulic parameters on the residence time (estimated as the site-average weighted calibrated mean radiocarbon age) and flux (Mg 1000 y$^{-1}$) per valley length of floodplain sediment. Predictor variables included contributing elevation ($z$), site-average depth of charcoal samples ($d$), channel slope ($S$), relative valley confinement ($C_v$; width of the channel ($w_c$) per width of the valley ($w_v$)), stream power ($\Omega = \gamma QS$, where $\gamma$ is the specific weight of water), and unit stream power ($\omega = \Omega$ $w_c^{-1}$).

Variable selection was conducted to reduce collinearity and achieve two observations per predictor for multiple linear regression analyses. Variables for multiple linear regression were selected iteratively by starting with the variable most strongly correlated with the outcome variable of interest, eliminating cross-correlated variables (Pearson's $r > 0.7$ and $p < 0.1$)[65], and then moving consecutively to the next most strongly correlated variable. Multiple linear regression analysis was conducted on the seven field study sites to examine controls on the residence time, flux of floodplain sediment (without the influence of recent fire) and net volume of floodplain sediment deposited along 155 study reaches in the 2013 flood.

Variables considered for the three multiple regression analyses included drainage area, elevation, valley confinement, channel slope, stream power, and unit stream power. Mean sample depth ($d$) was included as an additional variable in the analysis of radiocarbon ages at the eight study sites. Cumulative downstream distance, the downstream change in stream power relative to the adjacent upstream reach, the downstream change in unit stream power, the downstream change in valley confinement, and the net difference between sediment inputs and sediment outputs along the upstream reach were also considered for multiple linear regression analysis of the DoD data set. Potential predictor variables were included in a stepwise multiple linear regression analysis for the difference in sediment inputs and outputs along each reach (net sediment deposition), which included drainage area, valley confinement, channel slope, and the net difference in sediment inputs and outputs at the adjacent upstream study reach (net upstream sediment deposition).

Necessary transformations were made to meet the assumptions of multiple linear regression. Boxcox power transformation[66] of the mean age ($\lambda = -1.7172$), sediment flux ($\lambda = -0.020202$), and net volume of floodplain disturbed during the 2013 flood ($\lambda = 0.1576335$) was defined using the *boxcox* function of the MASS package in R[67]. The net volume of sediment deposited along each reach in the DoD analysis was added to 0.00001 before a power transformation. These transformations resulted in no significant departures from normality in model residuals or homoscedasticity through the failure to reject null hypotheses as determined by the Shapiro–Wilk normality test using the *Shapiro.test* function and the non-constant variance score test using the *ncvTest* function in R, respectively.

## Data availability

All supplementary figures and tables referenced in this manuscript are provided in the supplementary info file associated with this manuscript. Additional source data not provided in the manuscript are available on FigShare, an open-source data repository at https://doi.org/10.6084/m9.figshare.7849307.v1. Source data for figures are provided as a Source Data file for Figs. 1, 2, 5, 6, and 7, Supplementary Fig. 1 and 2, Supplementary Tables 3 and 4. Both pre- and post-flood lidar data that support the findings of this study are available in Colorado GeoData Cache at https://geodata.co.gov/.

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

## Acknowledgements

This paper is based upon work supported by National Science Foundation Grant No. DGE-0966346 "I-WATER: Integrated Water, Atmosphere, Ecosystems Education and Research Program" at Colorado State University (CSU); NSF Doctoral Research Dissertation Improvement Grant #1536186; and graduate student research grants from the Geological Society of America and the Colorado State University Warner College of Natural Resources and Department of Geosciences. We thank Michael Poulos for his insight on $^{14}$C age calibration, Georgie Bennett for sharing the DEM of difference and providing comments on versions of this manuscript, Teagan Deeney and Jim Self at the CSU Soil and Water Laboratory, and many people for their assistance in the laboratory and the field. This manuscript was greatly improved by comments from Grant Meyer and an anonymous reviewer.

## Author contributions

N.A.S. designed the study with the guidance of E.W.; N.A.S. coordinated and conducted all fieldwork and laboratory work with contributions from field assistants and E.W.; N.A. S. wrote the first draft and revised versions of the manuscript; E.W. provided comments and contributed to manuscript revisions.

## Additional information

**Competing interests:** The authors declare no competing interests.

