## [Peer Review File · Nature Communications]

Reviewers' comments:

Reviewer #1 (Remarks to the Author):

Review of Sutfin and Wohl, Floodplain sediment retention along mountain rivers altered by hydrologic and fire regimes, submitted to Nature Communications, NCOMMS-18-03847-T

Grant Meyer, University of New Mexico, gmeyer@unm.edu

March 5, 2018

Summary comments:

I'm grateful for the opportunity to review this interesting paper, which presents new information on floodplain sediment residence times along a mountain river, of interest to a broad range of scientists seeking to understand climate-related processes and mass transfers in mountain environments. Prior data of this sort are sparse, though timescales of sediment storage and evacuation are important to concerns such as water quality, reservoir sedimentation, and riverine ecosystems. An impressive amount of field work and data analysis went into this paper -- by their nature, such studies require a very determined effort in the field! The finding that residence times are longer in higher-elevation, glaciated valley reaches compared to lower unglaciated reaches is consistent with long-standing qualitative observations, where glacial valley widening and local gradient reduction (and overdeepening) provide greater accommodation space for sediment storage, and where terminal moraines provide strong downstream base level control for sediment accumulation. Unlike prior studies that are focused primarily on the chronology of change in fluvial systems, e.g. the timing of aggradation, degradation, this study uses a broad spatial sampling scheme for sediment depths and ^{14}C ages to provide quantitative information on fluvial sediment residence times and mass flux.

While the inference of longer sediment residence times in the broader glacial valley above the terminal moraine makes sense, the effect of greater flood magnitudes at lower elevations -- which is well documented in prior work in the Colorado Front Range (CFR) -- is difficult to separate from the effect of lesser sediment storage space in the confined lower valleys. While the hydroclimatic hypothesis makes sense, the presence of only two transects below the terminal moraine and the substantial scatter in ages within transects in general lessen support for the interpretation of a hydroclimatic control. Given the major effort already expended, I'm reluctant to suggest that more transects and samples are needed, e.g. providing larger sample populations and extending to lower elevations and higher order parts of the system, but without additional data, the interpretations need to be presented with appropriate caution.

The influence of changing fire regimes as highlighted in the title is more speculative, since there is little support for fire regime change in itself, at least in the data as presented. The intro to the paper states that the study sites span a range of wildfire regimes, but these different regimes are not described or related to differing sediment transport and storage processes. There is some reference to changing frequencies of fire over time, but those changes are not clearly documented (e.g. comments re. lines 201-203 below).

Overall, the writing could benefit from a more straightforward and explicit presentation of information, as explained in comments in the PDF. Also, while details of the research methods and protocols can be presented in the Methods section, a clear and consolidated explanation of the main research rationale and approach in the general introduction section will help greatly. There is also a bit of a tendency to extrapolate well beyond the study area, e.g. to recently deglaciated areas in the Conclusions, and the title itself refers to mountain rivers (rather than a mountain river); mountain rivers span a very broad range of climatic and tectonic environments and biomes, and this study does not apply to all.

Specific comments -- see also comments on the annotated ms. PDF:

My primary methodological concern with the paper is that the ^{14}C dating rationale is not clearly explained and justified. For estimating sediment residence time, the fundamental goal (as I understand it) would be to date the time of deposition of the floodplain sediment at many locations. The authors have generated a substantial dataset of ^{14}C ages on wood charcoal fragments, but how those ages are considered to relate to sediment depositional age is not well explained. Below are some possibilities and associated potential errors (in addition to the ^{14}C analytical uncertainty). The authors make some mention of most of these, but in detail the rationale is unclear.

1. **The charcoal was generated in a wildfire and underwent fluvial transport and deposition during the several-year period of postfire instability following.** Main source of error is age of wood > age of fire (inbuilt age of Gavin 2001), which could be up to a few hundred years for wood from the center of an old-growth tree, and probably less for dead wood at the time of fire in the CFR environment. However, most wildfire charcoal is produced on young outer wood, so the error is not likely to reach this maximum. This is the error of concern for dating wildfires, regardless of charcoal sample context, and it is recognized to some extent on lines 430-434 -- but the wording also seems to suggest that this error is not of concern for floodplain sediment residence time, which does not follow. It would apply to dating both the wildfire and sediment deposition time, and can add to error in the cases below.
2. **The charcoal was eroded from storage in hillslope soil or colluvium, tributary fans, or upstream channel or floodplain deposits, and redeposited at the floodplain sample site.** In this case charcoal age is potentially >> age of deposition, by 100s to 1000s of years. This reworked charcoal problem was investigated by Blong and Gillespie (1978, also cited by the authors) in an Australian river with upstream drainage area ~2000 km², where charcoal from the modern channel deposits gave ages up to ~1500 ^{14}C yr BP for the finest particles, and the coarsest (>4 mm), ~600 ^{14}C yr BP. Not noted by Blong and Gillespie (1978), however, is that the sampled Eucalyptus charcoal is from very hard and rot-resistant wood, thus much more durable and persistent in fluvial transport than the conifer softwood charcoal of the CFR study area, and transport distances are lower, such that the overall error is likely less than in the Australian case. CFR conifer

wood also rots relatively quickly and is unlikely to produce a large dead wood error (as in #1). Regardless, without more detailed sedimentologic and stratigraphic observations than can be obtained by augering, it is difficult to differentiate between postfire charcoal of type #1 and reworked, fluvially deposited charcoal. The above concerns are to some extent addressed on lines 454-457, which state “Because radiocarbon dating of charcoal essentially provides an age of the wildfire, this approach assumes that charcoal is incorporated into the floodplain soon after it burns. Thus, older pieces of charcoal that burned at higher elevations and have taken hundreds of years to travel downstream are increasingly likely to be incorporated into floodplain soils with increasing drainage area. Because of this confounding error and potential for long-tailed distributions in floodplain radiocarbon ages, estimates are interpreted as a mean maximum residence time of floodplain sediment”. Given the authors’ stated processes above, however, reworked charcoal ages are more accurately providing a residence time within the entire system above the sampling site, rather than at that site, and considering the residence time estimates as maxima does make sense if the charcoal is fluvially transported. Selecting the coarsest and most angular fragments for dating would help reduce this error, as charcoal is fragmented and rounded in fluvial transport, but charcoal is also likely to be broken during augering, making this more difficult. But what if the some of the dated charcoal was not fluvially transported, but produced at the sample site, as in #3 below?

- 3. The charcoal was produced by fire on the floodplain at its site of sampling (in situ),** and its age is younger than the underlying sediment, possibly by 100s-1000s yr; the age of the overlying deposits could also be younger by up to 100s-1000s yr. A charred soil surface layer that is well preserved and little bioturbated is good evidence that the overlying sediment was deposited soon after fire, but this requires close examination of intact stratigraphy, as such layers are typically < 1 cm thick and composed predominantly of fine charred litter (Meyer et al. 1995), likely to be disrupted and difficult to recognize in auger samples. Lines 448-449 state that “Augering of floodplain sediment revealed no evidence of a system-wide or even localized wildfire across individual study sites”, but it’s not explained what that evidence was expected to be. It is also noted on lines 123-124 that “horizontal discontinuity of ages provides no evidence of localized fire within undisturbed floodplain stratigraphy.” That is true, but it is not evidence against localized fire (meaning fire at the sample transect); charred soil surfaces are only locally formed, severe burns leave mostly ash rather than charred material, and even minor scour on a floodplain surface can result in discontinuous preservation of those layers.

So without more detailed examination of intact stratigraphy in addition to augering, it is difficult to say whether the sampled charcoal was produced in situ or not. Given that Sibold et al. (2006) mapped wildfires across many of the floodplain study sites, it is unwarranted to assume that in situ charcoal is not present. This also brings up the concern that vegetation at the study transects is not at all described in the ms., and this needs to be done concisely at least. A quick look with Google Earth indicates that all of

the study sites except AB4 appear conifer-forested, and these very likely burned repeatedly over the timescales of this study. Overall it is hard to know how much and in which direction this error would influence the results.

The above is not to suggest that the dating is so affected by errors as to invalidate the results, but the dating rationale and potential errors need to be clearly explained in the introduction to the study, and better considered as to their impact on interpretations. As is, the rationale is scattered throughout, including within the final Methods section, which should be reserved for the details of procedures. It is unfortunate that detailed sedimentologic and stratigraphic information is unavailable, given that in situ charred layers, inset relationships and buttress unconformities, buried A horizons, etc. would allow better assessment of ages. The most closely analogous study that I'm familiar with is the Lancaster et al. (2010) work on sediment residence times at tributary confluences in the Oregon Coast Range, where ^{14}C sampling paid close attention to stratigraphy and rejected clearly inverted ages. However, sampling by auger is understandable in a practical sense given that direct observation of stratigraphy would either be limited to bank exposures (and thus bias floodplain sediment ages), or require large amounts of time and excavation that may not be permitted in a national park.

Calibrated pooled mean ages along each study transect are used for interpretation. Pooled mean ages use the probability distribution for each age, and are usually employed where multiple ages have been obtained for same event, where the assumption is that the true age falls within a narrow time range. In contrast, here they describe the mean of a broad distribution of ages for a number of independent events. The pooled mean ages are given with a +/- of between 2 and 20 yr (Table 1 and Fig 2). I haven't looked in the CALIB code to see how specifically these uncertainties are calculated, but it would be more realistic to use a measure that better describes the distribution of ages. For example, 1 standard deviation of the weighted mean calibrated ages at AB1 = 1843 yr (compared to the pooled mean +/- of 4 yr).

No information is provided on the methods of the Sibold et al. (2006) fire history, but that is important to include. In Fig 4, comparison is made between the Sibold wildfire dates in the last ~350 yr and summed probability ^{14}C distributions from this study, but I'm not sure what inferences are being drawn, and there appear to be major inconsistencies; see comments in the PDF. There is a problem inherent in such comparisons that stems from the very large uncertainties in radiocarbon calibration over the last 400 yr (largely from fossil fuel dilution and bomb enrichment of atmospheric ^{14}C). Other studies that note the large uncertainties and multiple possible correlations between tree-ring dated wildfires and ^{14}C -dated fire-related sedimentation include Bigio et al. (2010, Fig. 6) and Frechette and Meyer (2009 Fig. 3).

With regards to the fire history, lines 130-131 note a recent series of wildfires, but it isn't made clear when or where these occurred. Later in the paragraph the "recently burned" site PC2 (burned 1978 on Fig 1) is excluded from the pooled mean age-elevation regression, but how is this expected to greatly alter the age distribution at this site, where others have also experienced fire in the last few hundred years? The young pooled mean age at PC2 could just

stem from the small sample size ($n = 3$). There is also an increased frequency of fire noted on lines 201-203 without defining the period of increased frequency, or clearly explaining the evidence for this -- it also needs to be explained what period of lower frequency exists to compare this to, and what specific data support these different frequencies. Fire regimes are emphasized in the title of the paper, but as regimes involve the frequency, severity, and areal extent of wildfires, there needs to be more information on those regimes both over space and time throughout the study area to clearly bring this component of the system into the interpretations.

References cited here and in annotated ms. (other than those cited in the reviewed ms.):

Bigio, E., Swetnam, T.W. and Baisan, C.H., 2010. A comparison and integration of tree-ring and alluvial records of fire history at the Missionary Ridge Fire, Durango, Colorado, USA. *The Holocene*, 20(7), pp.1047-1061.

Colman, S.M., Pierce, K.L. and Birkeland, P.W., 1987. Suggested terminology for Quaternary dating methods. *Quaternary Research*, 28(2), pp.314-319.

Frechette, J.D. and Meyer, G.A., 2009. Holocene fire-related alluvial-fan deposition and climate in ponderosa pine and mixed-conifer forests, Sacramento Mountains, New Mexico, USA. *The Holocene*, 19(4), pp.639-651.

Gavin, D.G., 2001. Estimation of inbuilt age in radiocarbon ages of soil charcoal for fire history studies. *Radiocarbon*, 43(1), pp.27-44.

Larsen, I.J., MacDonald, L.H., Brown, E., Rough, D., Welsh, M.J., Pietraszek, J.H., Libohova, Z., de Dios Benavides-Solorio, J. and Schaffrath, K., 2009. Causes of post-fire runoff and erosion: water repellency, cover, or soil sealing?. *Soil Science Society of America Journal*, 73(4), pp.1393-1407.

Wicherski, W., Dethier, D.P. and Ouimet, W.B., 2017. Erosion and channel changes due to extreme flooding in the Fourmile Creek catchment, Colorado. *Geomorphology*, 294, pp.87-98.

Reviewer #2 (Remarks to the Author):

Review of Sutfin and Wohl

General comments and overview

This work uses an analysis of radiocarbon ages (pooled means) from 8 watersheds in varying elevation, drainage area, degree of confinement, etc. in the Rockies, largely within Rocky Mountain National Park and an analysis of pre- and post-flood LiDAR DEMs to demonstrate that climate, fire history, geologic setting (historic Pleistocene terminal moraine) all act to control sediment residence times in floodplains of mountain rivers. The claims of the paper are not fundamentally surprising or novel – the finding matches the expectation that broader valleys would accumulate more sediment and result in longer sediment residence times. The interesting facet to the result is that this location occurs at higher elevations than one might hypothesize, but it corresponds to the glacial activity, which is ultimately explicable. However, other work is cited within the paper (Montgomery, 1999) that indicates spatial heterogeneity in mountain streams is an important driver in patterns of erosion and deposition within these landscapes. So it is difficult to assess how novel the finding is the way it is presented.

The work is publishable in Nature Communications, but the following concerns outlined within the review should be addressed before being considered. I recommend acceptance with revision according to the comments outlined here.

The work is timely and relevant to the field and as a result will be of high interest. Transport of fine sediment through fluvial networks is a high priority topic in geomorphology for a wide variety of reasons, but there are important limitations in the work that need to be addressed. There are a series of assumptions one has to make to equate sediment residence times with sediment age. Some of these are outlined in work not included here (Bolin and Rhode, 1973), but I believe are covered in others cited (Lancaster and Casebeer, 2007). It would be helpful to have a list of these assumptions, whether or not they are met, and what the consequences are for not meeting them (as I believe one significant one is not met in this study design): a closed system, but reservoirs exchange; steady state conditions (this is the one that is violated, in my opinion), and these are typically applied to a distribution of ages, not pooled averages. It is also worth explaining why the framework presented by Lancaster and Casebeer (2007) which has subsequently been built upon by others (Bradley and Tucker, Skalak and Pizzuto, Pizzuto et al) is not used. Why pooled averages instead of age distributions?

In addition, the discussion regarding changes in elevation is often mixed and interchanged with a discussion of processes that have been introduced as being specific to certain elevation ranges. That is, the montane zone is below 2300 m (as I understand it) and the CFR is below 2500 m but there are very few if any sites that fall in that range. It is clear that there is an elevation gradient, but not completely clear that it lines up enough with these zones to infer that these processes are the ones driving the patterns. At the end of the paper, there is a brief mention of the fact that these elevations could vary, but it seems like an afterthought.

The conceptual figure seems to suggest that high elevation corresponds with wider valleys (and this matches the discussion of the subalpine zone), but Figure 3 suggests that the pattern within single watersheds moving downstream is a little more noisy. Is there a distribution of valley widths with elevations for the other watersheds to demonstrate that this is true as a whole?

Although the writing is generally concise and clear, I struggled a bit with the organization and flow. I found myself constantly going back and forth in between sections to remind myself of certain things. I think this could be helped with a table that includes additional information about whether you classify the site as subalpine or montane and whether its in the CFR and so forth.

Specific comments:

Line 14 – add “relatively” before little is known

Line 16 – channel margins is a potentially confusing term. For those focused on in stream habitat, margins refer to the nearbank regions of the channel, not the valley itself. Floodplain is a more apt term.

Line 17 – the importance of the flood to the long-term radiocarbon results and study is not clear. I understand its significance within the context of the LiDAR differencing, but it’s not completely clear why it impacts the long-term results. Did the valley change that significantly as a result of the flood?

Line 23 – We show that lower stream power in broader glacial valleys increases sediment residence time. As mentioned, this is not necessarily a surprising result.

Lines 27 and 28 – I do not think this is categorically true and everywhere and a more nuanced read is appropriate (see Slater et al, 2015; Archfield et al, 2016).

Lines 60 to 62 – There are a series of assumptions required to use ages as a proxy for residence times. I think a discussion of these is warranted.

Line 65-66 – This justification is not clear to me. Why? How? (much older ages within floodplain stratigraphy can provide insight..)

Figure 1 – It would be helpful to delineate the critical elevations that you mention in the Introduction (or at least include a discussion of its inherent variability or shift as mentioned in the Discussion and Conclusions).

Lines 83 – 84 – Is 2300 m also the difference between subalpine and montane or is this a different elevation? Might be helpful to explain that this might be variable and why.

Line 93 – the way this is written it sounds as though it may not necessarily differ from what is expected by some.

Lines 96 to 101 – The factors exacerbating erosion and promoting storage seem to all line up in such a way as to favor storage in the subalpine zone. While this has been outlined for climate and geology, it should also be written to reflect the fire frequency or maybe a table should be included to list all the factors in each zone and whether they promote erosion or accumulation.

Table 1 – somewhere it should be clearly noted which sites are considered within the CFR, which sites are considered subalpine, and so forth.

Line 104 – How were the 8 sites selected and why weren’t there more study sites within the montane or below the CFR? It appears you only have one site right on the border of where you anticipate the hydroclimatic shift.

Line 260 – Is there really a high potential for old charcoal to be stored in downstream floodplains in areas with there is not much storage, as your conclusions suggest? The probability seems that it would be much reduced, if at all. In addition, it’s possibly like finding a needle in a haystack in terms of sampling, but I could have that incorrect.

Line 268 to 273 – If there is a way to introduce this idea sooner, it would really help contextualize your results.

Other questions for referees to consider

For manuscripts that may merit further consideration, it is also helpful if referees can advise on the following points:

NCOMMS-1803847A

Response to Reviewers

*We appreciate the supportive comments and constructive criticism by the reviewers. We have **responded to the listed reviewer comments below in blue italics**. We have made all the suggested changes and addressed the comments on the pdf by reviewer 1. Please let me know if you prefer us to copy and address each of those comments from the pdf.*

We have addressed the organization and focused the primary message of the paper. Based on comments from reviewers 1 and 2, we have conducted the statistical analysis again using mean ages rather than pooled ages. This provided a basis to clarify uncertainty in ages. We have clarified the timing of recorded fire history as well as the methods used to establish that history. We have also more thoroughly discussed the assumptions of our approach as suggested by reviewers 1 and 2. Additionally, we have considered past work, conducted additional analysis, and placed our work in the context of others regarding transit time distributions, as suggested by reviewer 2.

We feel these changes and additions have much improved the quality of the manuscript. We hope the reviewers find these changes to be satisfactory in the improvement of the quality of our manuscript.

*Sincerely,
Nicholas Sutfin*

Reviewers' comments:

Reviewer #1 (Remarks to the Author):

See comments submitted as a PDF file. Also submitted are the manuscript and supplemental information with comments attached to green highlights.

Grant Meyer

Review of Sutfin and Wohl, Floodplain sediment retention along mountain rivers altered by hydrologic and fire regimes, submitted to Nature Communications, NCOMMS-18-03847-T

Grant Meyer, University of New Mexico, gmeyer@unm.edu

March 5, 2018

Summary comments:

I'm grateful for the opportunity to review this interesting paper, which presents new information on floodplain sediment residence times along a mountain river, of interest to a broad range of scientists seeking to understand climate-related processes and mass transfers in mountain environments. Prior data of this sort are sparse, though timescales of sediment storage and evacuation are important to concerns such as water quality, reservoir sedimentation, and riverine ecosystems. An impressive amount of field work and data analysis went into this paper -- by their nature, such studies require a very determined effort in the field! The finding that residence times are longer in higher-elevation, glaciated valley reaches compared to lower unglaciated reaches is consistent with long-standing qualitative observations, where glacial valley widening and local gradient reduction (and overdeepening) provide greater accommodation space for sediment storage, and where terminal moraines provide strong downstream base level control for sediment accumulation. Unlike prior studies that are focused primarily on the chronology of change in fluvial systems, e.g. the timing of aggradation, degradation, this study uses a broad spatial sampling scheme for sediment depths and ¹⁴C ages to provide quantitative information on fluvial sediment residence times and mass flux.

We appreciate the supportive comments and recognition of our effort.

While the inference of longer sediment residence times in the broader glacial valley above the terminal moraine makes sense, the effect of greater flood magnitudes at lower elevations -- which is well documented in prior work in the Colorado Front Range (CFR) -- is difficult to separate from the effect of lesser sediment storage space in the confined lower valleys. While the hydroclimatic hypothesis makes sense, the presence of only two transects below the terminal moraine and the substantial scatter in ages within transects in general lessen support for the interpretation of a hydroclimatic control. Given the major effort already expended, I'm reluctant to suggest that more transects and samples are needed, e.g. providing larger sample populations and extending to lower elevations and higher order parts of the system, but without additional data, the interpretations need to be presented with appropriate caution.

Although we have compared a number of study sites with varying degrees of valley confinement in an attempt to compare sites of similar confinement at high and low elevations, the reviewer makes a significant point. The lower elevation sites are indeed more confined than higher

Reviewers' comments:

elevation sites. Because we do not have additional study sites and data from lower elevations, we have removed emphasis in the title, text, and interpretation regarding the hydroclimatic disturbance regime. Instead, we have moved a small component of this topic to the discussion.

The influence of changing fire regimes as highlighted in the title is more speculative, since there is little support for fire regime change in itself, at least in the data as presented. The intro to the paper states that the study sites span a range of wildfire regimes, but these different regimes are not described or related to differing sediment transport and storage processes. There is some reference to changing frequencies of fire over time, but those changes are not clearly documented (e.g. comments re. lines 201-203 below).

We have clarified the language, providing more specific details regarding the location of each study site within the basin relative to ecotones and different fire regimes documented by cited research. We have also clarified language regarding fire regime, as we did not intend to suggest a change in fire regime through time, but rather historical differences across the spatial longitudinal gradient. Discussion has been added to propose linkages between documented differences in flood magnitude and differences in fire regime that can provide suggestion for the differences we see in this study regarding residence time. In addition to this clarification, we have more thoroughly qualified our findings as suggesting that anticipated differences in wildfire could influence floodplain stability.

Overall, the writing could benefit from a more straightforward and explicit presentation of information, as explained in comments in the PDF. Also, while details of the research methods and protocols can be presented in the Methods section, a clear and consolidated explanation of the main research rationale and approach in the general introduction section will help greatly. There is also a bit of a tendency to extrapolate well beyond the study area, e.g. to recently deglaciated areas in the Conclusions, and the title itself refers to mountain rivers (rather than a mountain river); mountain rivers span a very broad range of climatic and tectonic environments and biomes, and this study does not apply to all.

We have revised the manuscript with specific comments in the pdf as guidance and rearranged text to provide a more straightforward presentation. We have added a more thorough explanation of the rationale and approach in the introduction. We do not mean to extrapolate our findings to other mountain rivers, so have reworded the text to emphasize our intention to suggest that further investigation of other mountain rivers is needed to examine similar relationships in similar glaciated mountain systems.

Specific comments -- see also comments on the annotated ms. PDF:

My primary methodological concern with the paper is that the 14C dating rationale is not clearly explained and justified. For estimating sediment residence time, the fundamental goal (as I understand it) would be to date the time of deposition of the floodplain sediment at many locations. The authors have generated a substantial dataset of 14C ages on wood charcoal fragments, but how those ages are considered to relate to sediment depositional age is not well explained. Below are some possibilities and associated potential errors (in addition to the 14C analytical uncertainty). The authors make some mention of most of these, but in detail the rationale is unclear.

We greatly appreciate the suggestions and further details regarding potential possibilities and

concerns. We have modified the text to more thoroughly explain the rationale. We use radiocarbon ages as a proxy for the timing of deposition (with discussion of the potential error involved) with consideration that the age is a maximum estimate of the duration charcoal may have been present within each floodplain. We also address comments and a suggestion by the other reviewer to use cumulative probability functions, which are typically used (with great caution as we highlighted using citations of other work) to infer depositional episodes and timing of distinct events. Instead, we use an average maximum pooled-mean age as an estimate of the maximum residence time for each study site.

1. The charcoal was generated in a wildfire and underwent fluvial transport and deposition during the several-year period of postfire instability following. Main source of error is age of wood > age of fire (inbuilt age of Gavin 2001), which could be up to a few hundred years for wood from the center of an old-growth tree, and probably less for dead wood at the time of fire in the CFR environment. However, most wildfire charcoal is produced on young outer wood, so the error is not likely to reach this maximum. This is the error of concern for dating wildfires, regardless of charcoal sample context, and it is recognized to some extent on lines 430-434 -- but the wording also seems to suggest that this error is not of concern for floodplain sediment residence time, which does not follow. It would apply to dating both the wildfire and sediment deposition time, and can add to error in the cases below.

We have added text to address potential for this error and cite the suggested reference.

2. The charcoal was eroded from storage in hillslope soil or colluvium, tributary fans, or upstream channel or floodplain deposits, and redeposited at the floodplain sample site. In this case charcoal age is potentially >> age of deposition, by 100s to 1000s of years. This reworked charcoal problem was investigated by Blong and Gillespie (1978, also cited by the authors) in an Australian river with upstream drainage area ~2000 km², where charcoal from the modern channel deposits gave ages up to ~1500 14C yr BP for the finest particles, and the coarsest (>4 mm), ~600 14C yr BP. Not noted by Blong and Gillespie (1978), however, is that the sampled Eucalyptus charcoal is from very hard and rot-resistant wood, thus much more durable and persistent in fluvial transport than the conifer softwood charcoal of the CFR study area, and transport distances are lower, such that the overall error is likely less than in the Australian case. CFR conifer wood also rots relatively quickly and is unlikely to produce a large dead wood error (as in #1). Regardless, without more detailed sedimentologic and stratigraphic observations than can be obtained by augering, it is difficult to differentiate between postfire charcoal of type #1 and reworked, fluvially deposited charcoal. The above concerns are to some extent addressed on lines 454-457, which state "Because radiocarbon dating of charcoal essentially provides an age of the wildfire, this approach assumes that charcoal is incorporated into the floodplain soon after it burns. Thus, older pieces of charcoal that burned at higher elevations and have taken hundreds of years to travel downstream are increasingly likely to be incorporated into floodplain soils with increasing drainage area. Because of this confounding error and potential for long-tailed distributions in floodplain radiocarbon ages, estimates are interpreted as a mean maximum residence time of floodplain sediment". Given the authors' stated processes above, however, reworked charcoal ages are more accurately providing a residence time within the entire system above the sampling site, rather than at that site, and considering the residence time estimates as maxima does make sense if the charcoal is fluvially transported. Selecting the coarsest and most angular fragments for dating would help reduce this error, as charcoal is fragmented and rounded in fluvial transport, but charcoal is also likely to be broken during

augering, making this more difficult. But what if the some of the dated charcoal was not fluvially transported, but produced at the sample site, as in #3 below?

We very much appreciate these comments and thoughts. It is true that the conifer wood in the study area is likely to be more easily broken and decomposed than that of the Blong and Gillespie (1978) study. We would like to assume that the charcoal dated in our study indicates the exact date of a wildfire that produced the sample in situ because that would provide us with a chronology of deposition and more certainty in our sediment residence times. However, we are careful not to assume that the potential error of reworked charcoal is not significant in this case. We ask the reviewer to please consider that although the conifer wood may be more easily decomposed than the eucalyptus wood in the Australian study, our study site spans only 200 km² whereas the Blong and Gillespie (1978) study was conducted within a basin an order of magnitude greater (i.e., 2000 km²). Traveling the greater of the two distances would be unlikely for the soft wood in our study, but we think it is appropriate to keep the potential for reworked charcoal in consideration in this size of a watershed. This is particularly important because the floodplain sediment contains evidence of sorting and fluvial transport.

3. The charcoal was produced by fire on the floodplain at its site of sampling (in situ), and its age is younger than the underlying sediment, possibly by 100s-1000s yr; the age of the overlying deposits could also be younger by up to 100s-1000s yr. A charred soil surface layer that is well preserved and little bioturbated is good evidence that the overlying sediment was deposited soon after fire, but this requires close examination of intact stratigraphy, as such layers are typically < 1 cm thick and composed predominantly of fine charred litter (Meyer et al. 1995), likely to be disrupted and difficult to recognize in auger samples. Lines 448-449 state that “Augering of floodplain sediment revealed no evidence of a system-wide or even localized wildfire across individual study sites”, but it’s not explained what that evidence was expected to be. It is also noted on lines 123-124 that “horizontal discontinuity of ages provides no evidence of localized fire within undisturbed floodplain stratigraphy.” That is true, but it is not evidence against localized fire (meaning fire at the sample transect); charred soil surfaces are only locally formed, severe burns leave mostly ash rather than charred material, and even minor scour on a floodplain surface can result in discontinuous preservation of those layers.

We have clarified our language and are now more careful with the inferences we make, more clearly stating the limitations of our approach.

So without more detailed examination of intact stratigraphy in addition to augering, it is difficult to say whether the sampled charcoal was produced in situ or not. Given that Sibold et al. (2006) mapped wildfires across many of the floodplain study sites, it is unwarranted to assume that in situ charcoal is not present. This also brings up the concern that vegetation at the study transects is not at all described in the ms., and this needs to be done concisely at least. A quick look with Google Earth indicates that all of the study sites except AB4 appear conifer-forested, and these very likely burned repeatedly over the timescales of this study. Overall it is hard to know how much and in which direction this error would influence the results.

We more thoroughly discussed the validity of the mapped fire history, correlation with ages of fires at higher elevations found in floodplains at lower elevations that did not experience the same fire, and explanation that charcoal was taken from floodplain sediment deposits. We have added description of the vegetation type along each study site (which each consist of 11

transects).

The above is not to suggest that the dating is so affected by errors as to invalidate the results, but the dating rationale and potential errors need to be clearly explained in the introduction to the study, and better considered as to their impact on interpretations. As is, the rationale is scattered throughout, including within the final Methods section, which should be reserved for the details of procedures. It is unfortunate that detailed sedimentologic and stratigraphic information is unavailable, given that in situ charred layers, inset relationships and buttress unconformities, buried A horizons, etc. would allow better assessment of ages. The most closely analogous study that I'm familiar with is the Lancaster et al. (2010) work on sediment residence times at tributary confluences in the Oregon Coast Range, where ^{14}C sampling paid close attention to stratigraphy and rejected clearly inverted ages. However, sampling by auger is understandable in a practical sense given that direct observation of stratigraphy would either be limited to bank exposures (and thus bias floodplain sediment ages), or require large amounts of time and excavation that may not be permitted in a national park.

We appreciate the comments and thoughts on this matter and also agree that more detailed stratigraphic information would be very beneficial.

Calibrated pooled mean ages along each study transect are used for interpretation. Pooled mean ages use the probability distribution for each age, and are usually employed where multiple ages have been obtained for same event, where the assumption is that the true age falls within a narrow time range. In contrast, here they describe the mean of a broad distribution of ages for a number of independent events. The pooled mean ages are given with a +/- of between 2 and 20 yr (Table 1 and Fig 2). I haven't looked in the CALIB code to see how specifically these uncertainties are calculated, but it would be more realistic to use a measure that better describes the distribution of ages. For example, 1 standard deviation of the weighted mean calibrated ages at AB1 = 1843 yr (compared to the pooled mean +/- of 4 yr).

We agree that the errors seemed to be unrealistically low and understand the reviewers point about the use of pooled mean. We have replaced the use of pooled mean ages with arithmetic means at each study reach and have represented standard deviations of those means on tables and figures.

No information is provided on the methods of the Sibold et al. (2006) fire history, but that is important to include. In Fig 4, comparison is made between the Sibold wildfire dates in the last ~350 yr and summed probability ^{14}C distributions from this study, but I'm not sure what inferences are being drawn, and there appear to be major inconsistencies; see comments in the PDF. There is a problem inherent in such comparisons that stems from the very large uncertainties in radiocarbon calibration over the last 400 yr (largely from fossil fuel dilution and bomb enrichment of atmospheric ^{14}C). Other studies that note the large uncertainties and multiple possible correlations between tree-ring dated wildfires and ^{14}C -dated fire-related sedimentation include Bigio et al. (2010, Fig. 6) and Frechette and Meyer (2009 Fig. 3).

We have added additional details regarding the methods of the Sibold et al. (2006) study and have clarified the noted inconsistencies as well as more appropriately qualifying our statements about the timing and comparison of specific fires in the record and ages we obtain in this study.

With regards to the fire history, lines 130-131 note a recent series of wildfires, but it isn't made clear when or where these occurred. Later in the paragraph the "recently burned" site PC2 (burned 1978 on Fig 1) is excluded from the pooled mean age-elevation regression, but how is this expected to greatly alter the age distribution at this site, where others have also experienced fire in the last few hundred years? The young pooled mean age at PC2 could just stem from the small sample size ($n = 3$). There is also an increased frequency of fire noted on lines 201-203 without defining the period of increased frequency, or clearly explaining the evidence for this -- it also needs to be explained what period of lower frequency exists to compare this to, and what specific data support these different frequencies. Fire regimes are emphasized in the title of the paper, but as regimes involve the frequency, severity, and areal extent of wildfires, there needs to be more information on those regimes both over space and time throughout the study area to clearly bring this component of the system into the interpretations.

We have more clearly identified periods and referenced the timing of specific fires we refer to and where we see potential corresponding ages with our data. We have also clarified our comments regarding more recent fires recorded in the record, explicitly identified peaks in our age records when referring to fires for comparison, added a new map insert to the summed probability figure to add in this comparison, changed the wording to identify specific fires rather than infer changes in frequency of fires, discussed how this might influence floodplain disturbance, altered the organization and wording regarding documented fire frequency and spatial patterns in the study area, and discussed potential bias based on a limited sample size.

References cited here and in annotated ms. (other than those cited in the reviewed ms.):

We appreciate the suggested references and have read and cited many of them.

Bigio, E., Swetnam, T.W. and Baisan, C.H., 2010. A comparison and integration of tree-ring and alluvial records of fire history at the Missionary Ridge Fire, Durango, Colorado, USA. *The Holocene*, 20(7), pp.1047-1061.

Colman, S.M., Pierce, K.L. and Birkeland, P.W., 1987. Suggested terminology for Quaternary dating methods. *Quaternary Research*, 28(2), pp.314-319.

Frechette, J.D. and Meyer, G.A., 2009. Holocene fire-related alluvial-fan deposition and climate in ponderosa pine and mixed-conifer forests, Sacramento Mountains, New Mexico, USA. *The Holocene*, 19(4), pp.639-651.

Gavin, D.G., 2001. Estimation of inbuilt age in radiocarbon ages of soil charcoal for fire history studies. *Radiocarbon*, 43(1), pp.27-44.

Larsen, I.J., MacDonald, L.H., Brown, E., Rough, D., Welsh, M.J., Pietraszek, J.H., Libohova, Z., de Dios Benavides-Solorio, J. and Schaffrath, K., 2009. Causes of post-fire runoff and erosion: water repellency, cover, or soil sealing?. *Soil Science Society of America Journal*, 73(4), pp.1393-1407.

Wicherski, W., Dethier, D.P. and Ouimet, W.B., 2017. Erosion and channel changes due to extreme flooding in the Fourmile Creek catchment, Colorado. *Geomorphology*, 294, pp.87-98.

Reviewer #2 (Remarks to the Author):

Review of Sutfin and Wohl

General comments and overview

This work uses an analysis of radiocarbon ages (pooled means) from 8 watersheds in varying elevation, drainage area, degree of confinement, etc. in the Rockies, largely within Rocky Mountain National Park and an analysis of pre- and post-flood LiDAR DEMs to demonstrate that climate, fire history, geologic setting (historic Pleistocene terminal moraine) all act to control sediment residence times in floodplains of mountain rivers. The claims of the paper are not fundamentally surprising or novel – the finding matches the expectation that broader valleys would accumulate more sediment and result in longer sediment residence times. The interesting facet to the result is that this location occurs at higher elevations than one might hypothesize, but it corresponds to the glacial activity, which is ultimately explicable. However, other work is cited within the paper (Montgomery, 1999) that indicates spatial heterogeneity in mountain streams is an important driver in patterns of erosion and deposition within these landscapes. So it is difficult to assess how novel the finding is the way it is presented.

We clarify any confusion or contradictions regarding our citations and interpretation of our data.

The work is publishable in Nature Communications, but the following concerns outlined within the review should be addressed before being considered. I recommend acceptance with revision according to the comments outlined here.

We appreciate this comment and have addressed the comments below and made the requested changes where noted.

The work is timely and relevant to the field and as a result will be of high interest. Transport of fine sediment through fluvial networks is a high priority topic in geomorphology for a wide variety of reasons, but there are important limitations in the work that need to be addressed. There are a series of assumptions one has to make to equate sediment residence times with sediment age. Some of these are outlined in work not included here (Bolin and Rhode, 1973), but I believe are covered in others cited (Lancaster and Casebeer, 2007). It would be helpful to have a list of these assumptions, whether or not they are met, and what the consequences are for not meeting them (as I believe one significant one is not met in this study design): a closed system, but reservoirs exchange; steady state conditions (this is the one that is violated, in my opinion), and these are typically applied to a distribution of ages, not pooled averages. It is also worth explaining why the framework presented by Lancaster and Casebeer (2007) which has subsequently been built upon by others (Bradley and Tucker, Skalak and Pizzuto, Pizzuto et al) is not used. Why pooled averages instead of age distributions?

We have created exceedance probability-transit time distribution plots and examine various fits to examine our data while citing references in the context of prior work and assumptions mentioned above.

In addition, the discussion regarding changes in elevation is often mixed and interchanged with

a discussion of processes that have been introduced as being specific to certain elevation ranges. That is, the montane zone is below 2300 m (as I understand it) and the CFR is below 2500 m but there are very few if any sites that fall in that range. It is clear that there is an elevation gradient, but not completely clear that it lines up enough with these zones to infer that these processes are the ones driving the patterns. At the end of the paper, there is a brief mention of the fact that these elevations could vary, but it seems like an afterthought.

We have added additional text regarding differences in proposed elevation boundaries in disturbance regime, placed our results into the context of biomes in addition to proposed elevation boundaries, and deemphasized conclusions that attribute these changes to specific processes. Instead, we have moved the conceptual figure and this portion to the discussion to contrast potential response to wildfires and lower elevation.

The conceptual figure seems to suggest that high elevation corresponds with wider valleys (and this matches the discussion of the subalpine zone), but Figure 3 suggests that the pattern within single watersheds moving downstream is a little more noisy. Is there a distribution of valley widths with elevations for the other watersheds to demonstrate that this is true as a whole?

We have clarified in the text that our study sites were selected along partially confined values and not confined valleys without floodplains. The figure caption has also been modified to explain that the valleys depicted represent study reaches where samples were collected but that longitudinal heterogeneity in valley confinement is present at both higher and lower elevations. We have added citation to a reference that examined differences in valley confinement along continuous river segments, which shows significant differences in valley confinement above and below the Pliocene terminal moraine in the study watershed.

Although the writing is generally concise and clear, I struggled a bit with the organization and flow. I found myself constantly going back and forth in between sections to remind myself of certain things. I think this could be helped with a table that includes additional information about whether you classify the site as subalpine or montane and whether its in the CFR and so forth.

We have rearranged the organization to obtain a more logical flow and improve readability. A column of Montane and Subalpine classification has also been added to the table of study sites and we have clarified language regarding these distinctions. We have also added an elevation contour line corresponding with the ecotone between montane and subalpine on Figure 1 and Figure 4 and conducted additional statistical analysis to test differences in between these biomes.

Specific comments:

Line 14 – add “relatively” before little is known

We have made this suggested edit

Line 16 – channel margins is a potentially confusing term. For those focused on in stream habitat, margins refer to the nearbank regions of the channel, not the valley itself. Floodplain is a more apt term.

We agree and have replaced channel margins with river corridors

Line 17 – the importance of the flood to the long-term radiocarbon results and study is not clear. I understand its significance within the context of the LiDAR differencing, but it's not completely clear why it impacts the long-term results. Did the valley change that significantly as a result of the flood?

We have emphasized the complete restructuring of floodplains as a result of the flood and have more clearly drawn a link between the lidar observations and long-term 14C results.

Line 23 – We show that lower stream power in broader glacial valleys increases sediment residence time. As mentioned, this is not necessarily a surprising result.

We have reworded this statement to more appropriately emphasize the significance of our results.

Lines 27 and 28 – I do not think this is categorically true and everywhere and a more nuanced read is appropriate (see Slater et al, 2015; Archfield et al, 2016).

We appreciate the references and have reworded the statement to emphasize our results in the context of our study area and the western US.

Lines 60 to 62 – There are a series of assumptions required to use ages as a proxy for residence times. I think a discussion of these is warranted.

We have added a discussion of these assumptions in the methods section and address some of them specifically with the additional analysis of transit time distributions.

Line 65-66 – This justification is not clear to me. Why? How? (much older ages within floodplain stratigraphy can provide insight..)

We have deleted this sentence and reworded our statement the significance of older ages in floodplains.

Figure 1 – It would be helpful to delineate the critical elevations that you mention in the Introduction (or at least include a discussion of its inherent variability or shift as mentioned in the Discussion and Conclusions).

We have added a contour line representing the ecotone, retain the contour representing the hydrogeomorphic elevation threshold proposed by others, and provide additional details in the caption.

Lines 83 – 84 – Is 2300 m also the difference between subalpine and montane or is this a different elevation? Might be helpful to explain that this might be variable and why.

We have clarified these two elevation thresholds in the figure and the text, as mentioned above.

Line 93 – the way this is written it sounds as though it may not necessarily differ from what is expected by some.

We have rephrased and clarified this sentence to emphasize that while contemporary geomorphic understanding recognizes heterogeneity in the presence of erosive and depositional reaches, our hypothesis differs from the traditional understanding that steep mountainous environments are erosive without much recognition that they could be sediment reservoirs.

Lines 96 to 101 – The factors exacerbating erosion and promoting storage seem to all line up in such a way as to favor storage in the subalpine zone. While this has been outlined for climate and geology, it should also be written to reflect the fire frequency or maybe a table should be included to list all the factors in each zone and whether they promote erosion or accumulation.

We appreciate this suggestion and have made edits to the text to emphasize this point in the text and in figure 6.

Table 1 – somewhere it should be clearly noted which sites are considered within the CFR, which sites are considered subalpine, and so forth.

We have edited the text to more clearly state that all sites are in the CFR and to characterize sites by biome, glaciated history, elevation, and wildfire frequency.

Line 104 – How were the 8 sites selected and why weren't there more study sites within the montane or below the CFR? It appears you only have one site right on the border of where you anticipate the hydroclimatic shift.

We have more explicitly stated how the study sites were selecting, why more sites at the transition are not included, and discuss difference in the montane and subalpine zones and the those above and below the Pleistocene terminal glacial moraine.

Line 260 – Is there really a high potential for old charcoal to be stored in downstream floodplains in areas with there is not much storage, as your conclusions suggest? The probability seems that it would be much reduced, if at all. In addition, it's possibly like finding a needle in a haystack in terms of sampling, but I could have that incorrect.

We have clarified the language in the methods the discussion to explain the increased probability of older charcoal with increasing drainage area. The probability of charcoal being incorporated into floodplain sediment after significant travel time before reaching the floodplain is much lower at small headwater basins compared to those at lower elevations with larger drainage areas in which charcoal could have traveled long distances before being incorporated into the floodplains. We also emphasized that finding otherwise increases support for our hypothesis that floodplains at lower elevations have shorter residence times compared to those higher elevations.

Line 268 to 273 – If there is a way to introduce this idea sooner, it would really help contextualize your results.

We have discussed the uncertainty in the elevation of this transitional boundary sooner in the “Mountain Streams and the Colorado Front Range” section and the discussion regarding the potential for shifting transitions over time.

Other questions for referees to consider

For manuscripts that may merit further consideration, it is also helpful if referees can advise on the following points:

Reviewers' comments:

Reviewer #1 (Remarks to the Author):

Review of Sutfin and Wohl revised ms., Sediment residence times along high-elevation glaciated mountain river corridors, submitted to Nature Communications, NCOMMS-18-03847A

Grant Meyer, University of New Mexico, gmeyer@unm.edu

30 August 2018

I'm also submitting these comments as a PDF file, along with an annotated ms. PDF (including the supplemental material) and a comment summary PDF of the main ms.

General comments:

This manuscript has been improved from the first submission, but important concerns remain, in my view most importantly in the treatment of the 14C age-residence time data, as explained in the numbered comments below.

The writing also needs further work; I have made a number of highlighted comments in the annotated PDF, and have also included a separate file of these comments. The introductory section needs a more logical organization and better clarity (see annotated PDF); for the Mountain Streams and the Colorado Front Range section starting line 85, I'm repeating a comment here: The beginning of this section is the place to make clear definitions of the montane, subalpine, and alpine zones, including their elevation ranges, geomorphic character, hydroclimate, and vegetation, in a concise fashion, with info on each zone together and treated in order - as in the first draft, this information remains scattered piecemeal throughout. You can then provide more detail on specifics of each study site as necessary, following.

Overall there are many instances where descriptions need to be more clear and complete. Adding clear topic sentences would help the reader understand the main point of paragraphs and sections.

1. The authors' respond that they replaced the use of pooled mean ages with arithmetic means at each study reach and have represented standard deviations of those means on tables and figures, however, reference to pooled means still remains in Table 2 and on lines 505 and 592. There are at least some other parts of the ms. that were missed in this revision, e.g. lines 246-247.

2. In the authors' response to my comment 2 in the list of 3 possible relationships between charcoal age and sediment age: The charcoal was eroded from storage ... and redeposited at the floodplain sample site – they state that “we think it is appropriate to keep the potential for reworked charcoal in consideration in this size of a watershed”. Of course, I completely agree – reworking of charcoal in this CO Front Range fluvial system is likely, and was the main point of my comment #2 on potential relations between charcoal age and sediment age. The probability of fluvial charcoal reworking is precisely why, in our work on 14C dating of fire-related sedimentation events, we sample alluvial-fan depositional sites directly below and proximal to hillslopes, where the potential for reworking is minimized, and use diagnostic sedimentologic and stratigraphic evidence to interpret fire-related deposits; we avoid fluvial floodplain deposits in constructing a fire-related event chronology (e.g. Meyer et al., 1995) Recognition of the potential for reworking was also re-emphasized at the end of my comment: “But what if the some of the dated charcoal was not fluvially transported, but produced at the sample site...?” (emphasis added; and see the next comment below). The point of my comments re. Blong and Gillespie (1978) was to emphasize that one cannot expect the same degree of charcoal preservation and reworking in the CO Front Range as in that Australian study, with smaller basin size and softwood charcoal in the former, as the authors note. This is worth concise discussion in the text.

3. Also relating to the 3 possibilities for charcoal age vs. sediment age, the authors respond that “We use radiocarbon ages as a proxy for the timing of deposition (with discussion of the potential error involved) with consideration that the age is a maximum estimate of the duration charcoal may have been present within each floodplain” (emphasis mine). But possibility 3 is that the charcoal was produced by fire on the floodplain at its site of sampling (in situ), and its age is younger than the underlying sediment, possibly by 100s-1000s yr. In that case the charcoal age is a minimum age for the underlying sediment. It’s a maximum age for overlying sediment, but neither age is necessarily a close limiting age.

4. Figure 2 and related text: I’m uncertain as to what was done in plotting residence times (from 14C ages) here. The Fig. 2 caption states that “Solid black circles indicate median ages for all samples peaks (96.5% confidence level) in each study area and error bars represent the standard deviations of those means.” The methods text on lines 508-510 also notes that “all peaks in each sample” were plotted, without explaining what is meant by sample peaks. It is clear that many more data points are plotted for each site than the 14C ages obtained there, e.g. site PC4 has eight 14C ages, but there are 15 points plotted on the figure. I’m guessing that “sample peaks” means peaks (i.e. modes) in the calibrated probability distribution for a 14C age, as these age-probability distributions are commonly multimodal. If this is the case, I don’t understand the rationale – what would justify turning a single age into several age (or residence time) data points, when the age represents a single time of sediment storage? Four of the 5 site have 6 to 9 ages, which is limited data to define a site age distribution, but those are the data to be worked with. At the very least the method and rationale here need to be fully and clearly explained, and justified if that is possible. Use of “all peaks for calibrated mean ages” is also mentioned in calculating standard deviation error bars in the Fig. 3 caption, and this also needs to be clearly explained and justified. Why not just use the mean and SD of the single weighted mean or median of the calibrated probability distribution for each single age?

Reviewer #2 (Remarks to the Author):

The revised manuscript is much improved. The authors clearly addressed all of the previous review comments. This work uses an analysis of radiocarbon ages from 8 watersheds in varying elevation, drainage area, degree of confinement, etc. in the Rockies, largely within Rocky Mountain National Park and an analysis of pre- and post-flood LiDAR DEMs to demonstrate that climate, fire history, geologic setting (historic Pleistocene terminal moraine) all act to control sediment residence times in floodplains of mountain rivers. The work is timely and relevant to the field and as a result will be of high interest. Transport of fine sediment through fluvial networks is a high priority topic in geomorphology for a wide variety of reasons. The text reorganization has greatly improved the clarity and readability. The more carefully worded statements regarding interpretation and discussion of assumptions are also a great improvement. The addition of the age distributions and their implications greatly add to the discussion of residence times and puts the work into context of the building literature on residence times. The modifications of the discussion of the conceptual model and its focus on elevation differences with implications for process is also a great improvement. Overall, the paper is greatly improved and I have no additional comments or changes to make. As is stands, it is accepted for publication in Nature Communications.

Response to Reviewers Comments: NCOMMS-18-03847A

*Below, we respond to the general comments in blue italics. In black text following the general comments by Reviewer #1, we have copied and pasted the comments from the annotated pdf by Reviewer #1 and listed the line numbers from the prior version of the manuscript for which these comments correspond. In blue italics beneath each of these comments, we list the corresponding **line numbers on the revised manuscript with tracked changes** followed by our response to each comment.*

We greatly appreciate the support and comments from this second round of reviews. We are thankful Reviewer #2 feels we have adequately addressed prior comments and that our manuscript is ready for acceptance and publication in Nature Communications. We have addressed all the comments by reviewer #1, Grant Meyer, and feel these comments have greatly improved the quality and presentation of our manuscript. In addition to our responses to general comments, we have also pasted comments from the annotated pdf and written responses to each of those comments below. We have reconducted a minor analysis based on Dr. Meyer's suggestion to create the exceedance probability plots using only the weighted mean ages rather than all the "peaks" (modes) in the probability distribution function. Although this reduced the number of points in each plot, this analysis resulted in the same overall outcome and interpretation as the prior version of this manuscript. We have also clarified language regarding our use of calibrated weighted mean radiocarbon ages and site-average calibrated weighted mean radiocarbon ages with elimination of all mention of pooled means, which was the original analysis used in the first version of this manuscript. We have clarified language and included more specific discussion of the assumptions and potential errors with our approach. As mentioned, these edits have greatly improved the manuscript. We have also provided additional details regarding field methods, bulk density calculations, and estimates of sediment volumes, which we realize should be included. We hope that these edits are satisfactory to Reviewer #1 and look forward to a response.

Reviewers' comments:

Reviewer #1 (Remarks to the Author):

Review of Sutfin and Wohl revised ms., Sediment residence times along high-elevation glaciated mountain river corridors, submitted to Nature Communications, NCOMMS-18-03847A

Grant Meyer, University of New Mexico, gmeyer@unm.edu

30 August 2018

I'm also submitting these comments as a PDF file, along with an annotated ms. PDF (including the supplemental material) and a comment summary PDF of the main ms.

General comments:

This manuscript has been improved from the first submission, but important concerns remain, in

my view most importantly in the treatment of the 14C age-residence time data, as explained in the numbered comments below.

As noted in our responses below, we have thoroughly addressed each of these comments and concerns in our analysis and the text.

The writing also needs further work; I have made a number of highlighted comments in the annotated PDF, and have also included a separate file of these comments. The introductory section needs a more logical organization and better clarity (see annotated PDF); for the Mountain Streams and the Colorado Front Range section starting line 85, I'm repeating a comment here: The beginning of this section is the place to make clear definitions of the montane, subalpine, and alpine zones, including their elevation ranges, geomorphic character, hydroclimate, and vegetation, in a concise fashion, with info on each zone together and treated in order - as in the first draft, this information remains scattered piecemeal throughout. You can then provide more detail on specifics of each study site as necessary, following.

We have made the suggested edits to the organization and writing, rearranging this referenced section specifically how the reviewer has suggested. We have also made changes in provided clarity in all other places the reviewer has suggested, and improved the writing in additional places as well. This paragraph appears on lines 193-211

Overall there are many instances where descriptions need to be more clear and complete. Adding clear topic sentences would help the reader understand the main point of paragraphs and sections.

We have conducted edits in numerous places to provide clarification and completeness of writing including reorganization of the ecoregions (mentioned above) and text in the discussion. We have provided more clear topic sentences in many cases and simplified complex or compound sentences.

1. The authors' respond that they replaced the use of pooled mean ages with arithmetic means at each study reach and have represented standard deviations of those means on tables and figures, however, reference to pooled means still remains in Table 2 and on lines 505 and 592. There are at least some other parts of the ms. that were missed in this revision, e.g. lines 246-247.

We have eliminated all references to pooled means and verified values in tables and text to match the site-average calibrated weighted mean radiocarbon ages. These changes have been made:

- *in Table 2, below line 576*
- *Line 1200*
- *Line 1319*

2. In the authors' response to my comment 2 in the list of 3 possible relationships between charcoal age and sediment age: The charcoal was eroded from storage ... and redeposited at the

floodplain sample site – they state that “we think it is appropriate to keep the potential for reworked charcoal in consideration in this size of a watershed”. Of course, I completely agree – reworking of charcoal in this CO Front Range fluvial system is likely, and was the main point of my comment #2 on potential relations between charcoal age and sediment age. The probability of fluvial charcoal reworking is precisely why, in our work on 14C dating of fire-related sedimentation events, we sample alluvial-fan depositional sites directly below and proximal to hillslopes, where the potential for reworking is minimized, and use diagnostic sedimentologic and stratigraphic evidence to interpret fire-related deposits; we avoid fluvial floodplain deposits in constructing a fire-related event chronology (e.g. Meyer et al., 1995) Recognition of the potential for reworking was also re-emphasized at the end of my comment: “But what if the some of the dated charcoal was not fluvially transported, but produced at the sample site...?” (emphasis added; and see the next comment below). The point of my comments re. Blong and Gillespie (1978) was to emphasize that one cannot expect the same degree of charcoal preservation and reworking in the CO Front Range as in that Australian study, with smaller basin size and softwood charcoal in the former, as the authors note. This is worth concise discussion in the text.

We appreciate the clarification and have reworded portions of the text, adding additional discussion about the reworking of charcoal and limitations the potential errors have on our interpretation of the results. We also make a qualitative comparison with that from Blong and Gillespie (1978) and provide additional discussion and citations for turnover times of wood in the study area.

These changes and clarifications are made on lines 488 to 514 and lines 1133 to 1136 of the new tracked changes manuscript.

3. Also relating to the 3 possibilities for charcoal age vs. sediment age, the authors respond that “We use radiocarbon ages as a proxy for the timing of deposition (with discussion of the potential error involved) with consideration that the age is a maximum estimate of the duration charcoal may have been present within each floodplain” (emphasis mine). But possibility 3 is that the charcoal was produced by fire on the floodplain at its site of sampling (in situ), and its age is younger than the underlying sediment, possibly by 100s-1000s yr. In that case the charcoal age is a minimum age for the underlying sediment. It’s a maximum age for overlying sediment, but neither age is necessarily a close limiting age.

As mentioned above, we have reworded portions of the text, adding additional discussion about in situ production of charcoal on the floodplain, how that can result in much younger mean ages, how this might provide a minimum age for underlying sediment. In discussing potential error and limitations in interpretation of constraints on both minimum and maximum ages of floodplain sediment deposition, we have also eliminated statements that our ages represent a minimum or maximum age for individual samples where appropriate. These changes were made on lines 112, 114-116, 520-521, 551, 1142-1206, and 1284.

4. Figure 2 and related text: I’m uncertain as to what was done in plotting residence times (from 14C ages) here. The Fig. 2 caption states that “Solid black circles indicate median ages for all samples peaks (96.5% confidence level) in each study area and error bars represent the standard deviations of those means.” The methods text on lines 508-510 also notes that “all peaks in each

sample” were plotted, without explaining what is meant by sample peaks. It is clear that many more data points are plotted for each site than the 14C ages obtained there, e.g. site PC4 has eight 14C ages, but there are 15 points plotted on the figure. I’m guessing that “sample peaks” means peaks (i.e. modes) in the calibrated probability distribution for a 14C age, as these age-probability distributions are commonly multimodal. If this is the case, I don’t understand the rationale – what would justify turning a single age into several age (or residence time) data points, when the age represents a single time of sediment storage? Four of the 5 site have 6 to 9 ages, which is limited data to define a site age distribution, but those are the data to be worked with. At the very least the method and rationale here need to be fully and clearly explained, and justified if that is possible. Use of “all peaks for calibrated mean ages” is also mentioned in calculating standard deviation error bars in the Fig. 3 caption, and this also needs to be clearly explained and justified. Why not just use the mean and SD of the single weighted mean or median of the calibrated probability distribution for each single age?

We have conducted a reanalysis of the data using the weighted means rather than all “peaks” for each sample and clarified this approach in the text. Because radiocarbon ages are best presented as a range and point estimates are not true representations of the probability distribution of 14C ages, particularly with multimodal pdfs as the reviewer notes, we first plotted cumulative modes with associated error for each sample. However, the reviewer’s point is well taken and we agree that normalizing the cumulative probability of the weighted means is a more intuitive and appropriate approach. In doing so, there are simply less points on our graph. We have however, chosen to plot the error bars to represent the range of ages at the 95.4% confidence level rather than the standard deviation because we feel it provides a more specific visualization of the data and uncertainty in the calibrated weighted mean ages. While R² and RMSE values are slightly different, the results and general interpretation of the analysis remain the same as they were in the prior version of the manuscript. The caption for Fig 3 was not corrected in the prior version of the manuscript since this plot used only the site-average calibrated weighted means for the points and the error bars.

These changes were made in Fig 2 below line 597, the caption for Fig 2 on line 622, the caption for Fig 3 on line 638. Refer to lines 1246-1259 for additional added details regarding these methods and the creation of the plots in Fig 2.

Below, please find the pasted comment from the annotated pdf with the appropriate reference to the line number in the prior version of the manuscript. Below each of these comments, we reference the line number in the pdf with tracked changes where related changes were made and provide our response to each comment in blue italics.

Line 38: Author: Owner Subject: Highlight Date: 8/29/2018 9:27:52 AM
down hillslopes would make more sense

Line 54: We have made this suggested edit

Line 47: Author: Owner Subject: Cross-Out Date: 8/29/2018 9:34:19 AM

Line 65: We have made this suggested edit

Line 63: Author: Owner Subject: Highlight Date: 8/29/2018 9:47:16 AM
mean of what specifically?

Lines 102-104: We have clarified this statement.

Line 69: Author: Owner Subject: Highlight Date: 8/29/2018 9:49:02 AM
this is not the only source of inbuilt age - see my 1st draft comments and Gavin 2001

Line 112 and lines 438-447, and lines 1040-1044: We have stated this more generally at first mention here and expand upon it in more detail to correctly identify sources of inbuilt ages

Line 75: Author: Owner Subject: Highlight Date: 8/29/2018 9:51:16 AM
again, mean of what specifically? all ages at a site, or transect?

Lines 107-111: We have clarified this statement.

Line 79: Author: Owner Subject: Highlight Date: 8/29/2018 9:53:13 AM
not necessarily maximum - see comments

Lines 112 and 1054-1102: We have deleted this reference to maximum floodplain sediment age and provide context for the results in the discussion of errors associated with our ages.

Line 85: Author: Owner Subject: Highlight Date: 8/29/2018 1:54:41 PM
The beginning of this section is the place to make clear definitions of the montane, subalpine, and alpine zones, including their elevation ranges, geomorphic character, hydroclimate, and vegetation, in a concise fashion, with info on each zone together and treated in order - as in the first draft, this information remains scattered piecemeal throughout. You can then provide more detail on specifics of each study site as necessary, following.

Lines 193-211: We have reorganized the requested information into a single paragraph in the manner suggested.

Line 87-88: Author: Owner Subject: Highlight Date: 8/29/2018 1:42:14 PM
The hydroclimatic shift isn't really driven by a threshold - the threshold is driven by meteorological changes with elevation.

Line 119: We have simplified and clarified this statement.

Line 91: Author: Owner Subject: Inserted Text Date: 8/29/2018 12:43:26 PM
Transport

Line 191: We have made this suggested edit.

Line 99: Author: Owner Subject: Highlight Date: 8/29/2018 12:49:13 PM

meaning, gives an example of? 155 reaches aren't shown - 155 cross sections, of which these are an example?

Lines 228-229, Fig. 1: We have clarified this statement in the caption.

Line 103-104: Owner Subject: Highlight Date: 8/29/2018 12:59:14 PM
As above, you haven't yet defined what these zones are

Lines 193-211: We have reorganized the requested information into a single paragraph in the manner suggested.

Line 107: Owner Subject: Highlight Date: 8/29/2018 1:00:50 PM
Wording implies that tributary channels are fan-shaped - instead they have built tributary fans

Line 201: We have clarified this statement and moved it up with the section that describes the vegetation and ecoregion.

Line 112-113: Author: Owner Subject: Highlight Date: 8/29/2018 1:11:40 PM
this is the way to state it - as in comment above, hydroclimate controls the threshold

Lines 230-232: Agreed, no changes have been made.

Line 124-129: Author: Owner Subject: Highlight Date: 8/29/2018 1:15:22 PM
unclear which zone is being described in terms of riparian veg - again, define discuss each zone separately

Lines 193-211: We have reorganized the requested information into a single paragraph in the manner suggested.

Line 131: Owner Subject: Highlight Date: 8/29/2018 1:15:44 PM
elevation?

Lines 253: We have deleted longitudinal and simplified the statement.

Line 134: Author: Owner Subject: Highlight Date: 8/29/2018 1:17:08 PM
Lake-sediment charcoal

Line 253: We modified this statement to read "lacustrine charcoal"

Line 135: Author: Owner Subject: Highlight Date: 8/29/2018 1:17:46 PM
meaning charcoal accumulation rate, or?

Line 255: Yes, we have made the suggested edit.

Line 137 - 138: Author: Owner Subject: Highlight Date: 8/29/2018 1:18:54 PM
these are times, not rates - clarify

Lines 257-258: We have clarified this statement

Line 139-141: Author: Owner Subject: Highlight Date: 8/29/2018 1:41:09 PM
Which sites are in subalpine vs. montane zones? Better here than in next paragraph, esp. if you have organized the zone description above

Lines 389-393: We have clarified this as suggested and also stated here which sites are above and below the Pleistocene terminal moraine.

Lines 147-149: Author: Owner Subject: Highlight Date: 8/29/2018 1:24:01 PM
run-on, awkward

Lines 408-409: We have revised and shortened this sentence.

Lines 149-150: Author: Owner Subject: Highlight Date: 8/29/2018 1:37:58 PM
shorten, simplify

Lines 411-427: We have shortened and simplified this sentence.

Line 150: Author: Owner Subject: Highlight Date: 8/29/2018 1:38:49 PM
reads as if erosion was "complete" - clarify

Line 410-411: We have clarified this statement.

Line 161: Author: Owner Subject: Highlight Date: 8/29/2018 1:55:56 PM
??

Lines 233-239: We have corrected and clarified this statement in our reorganization of the description of the study area and vegetation.

Line 162: Author: Owner Subject: Highlight Date: 8/29/2018 1:55:49 PM
lodgepole pine

Line 400: We have corrected this typo.

Line 177: Author: Owner Subject: Highlight Date: 8/29/2018 1:57:54 PM
,

Line 440: We have corrected this typo and make the suggested edit.

Lines 178-179: Author: Owner Subject: Highlight Date: 8/29/2018 2:01:52 PM
This is a very broad generalization and a bit of a straw man - it's not surprising or unexpected that there would be greater and longer-lasting sediment storage in lower-gradient reaches of broad glacial trough valleys, e.g. in stepped valley profiles or above moraine dams

Line 441: We have deleted this statement.

Line 186: Author: Owner Subject: Highlight Date: 8/30/2018 8:02:15 AM
Sediment, more accurately - most would have little pedogenesis and soil properties were not the point of Sampling

Line 478: We have made the suggested edit.

Line 188: Author: Owner Subject: Highlight Date: 8/29/2018 2:03:15 PM
??

Line 479: We have corrected and clarified this typo/sentence

Line 188: Author: Owner Subject: Highlight Date: 8/29/2018 2:03:03 PM
DirectAMS - explain that this is a commercial lab, could be mistaken for a technique

Line 480: We have made this suggested edit.

Line 189: Author: Owner Subject: Highlight Date: 8/29/2018 2:06:58 PM
better, but could clarify as "the mean of between 2 and 15 calibrated 14C ages obtained at each site..."

Line 481: We have restated this sentence similar to the suggested edit.

Line 193: Author: Owner Subject: Highlight Date: 8/29/2018 2:08:52 PM
more accurately, older inbuilt ages from inner wood of trees or charred dead wood

Lines 112 – 113: We used this phrasing to clarify.

Line 490: We have made similar specific statements.

Line 194: Author: Owner Subject: Highlight Date: 8/30/2018 9:00:59 AM

It is unclear what comment or suggestion was made here.

Line 194-195: Author: Owner Subject: Highlight Date: 8/29/2018 2:15:34 PM
Does floodplain mixing mean bioturbation, which is a definite possibility (e.g. burrowing, tree throw, ...)? Bioturbation should be stated as such. Also, in situ charcoal production doesn't in itself produce a stratigraphic age inversion, it just may substantially postdate the time of deposition of the underlying sediment, and predate overlying sediment, unless there is fire-related sedimentation or simply chance deposition shortly after fire. Erosion and deposition are part of reworking.

Lines 113-114: We have clarified this statement regarding bioturbation and added it much further up in the manuscript.

Lines 497-517 and 1137-1140: We have more thoroughly explained reworking of charcoal.

Lines 201-202: Author: Owner Subject: Highlight Date: 8/29/2018 3:18:41 PM
awk insert - break this sentence in two, first age range, then correlation with depth

Lines 540-542: We have made the suggested edit.

Line 220, Table 2: Author: Owner Subject: Highlight Date: 8/29/2018 3:25:47 PM
??

Table 2, below line 577: We have corrected the table.

Line 220, Table 2: Author: Owner Subject: Highlight Date: 8/29/2018 3:26:27 PM
not "pooled", correct?

Table 2, below line 577: Correct. We have corrected this typo in the table

Lines 222-223: Author: Owner Subject: Highlight Date: 8/30/2018 9:01:48 AM
see general comment - many more than the actual number of ages per site are plotted

Fig 2 below line 597, the caption for Fig 2 on line 622, the caption for Fig 3 on line 638: We have conducted analysis, as suggested, with only the mean ages rather than all peaks and made new plots where each point represents a mean age of an individual sample.

Line 238, Fig 2: Author: Owner Subject: Highlight Date: 8/30/2018 8:04:51 AM
unclear what was done here - see general comments

Lines 623 – 624: We have corrected this text in the caption. As mentioned above, we have conducted the analysis again using individual weighted mean ages and appropriately corrected the caption.

Line 243: Author: Owner Subject: Highlight Date: 8/30/2018 9:12:13 AM
Provide the evidence for this here, or at least explain that it is presented later

Lines 628-629: We state that we discuss this in detail later.

Lines 246-247: Author: Owner Subject: Highlight Date: 8/30/2018 9:07:53 AM
doesn't match Table 1, needs updating

Lines 630-631: We have updated this text to match Table 1.

Line 254, Fig 3: Author: Owner Subject: Highlight Date: 8/30/2018 8:06:05 AM
what is meant by peaks? see general comments

Line 239: We have corrected this statement to clarify that we use site averages of weighted mean ages as described above.

Line 278: Author: Owner Subject: Highlight Date: 8/30/2018 9:13:00 AM

How were these mapped?

Line 676: We have added text similar to that stated in the introduction, that fire history was reconstructed using dendrochronology and fire scars.

Line 281: Author: Owner Subject: Highlight Date: 8/30/2018 9:15:18 AM
meaning increased probability in the summed charcoal age distribution?

Line 679: Yes, we have clarified this statement.

Line 294, Fig 5: Author: Owner Subject: Highlight Date: 8/30/2018 9:27:46 AM
explain more fully - what type of sample?

Line 705: We have clarified this statement.

Lines 306-307: Author: Owner Subject: Highlight Date: 8/30/2018 9:34:48 AM
longer than what?

Line 722: We have clarified this statement.

Line 320: Author: Owner Subject: Highlight Date: 8/30/2018 9:36:45 AM see general comments
- how confidently can this be done with 6-9 14C ages?

Lines 733-739: We have added a couple of sentences about the limitations of our sample size and what it means in the context of our results.

Line 332, Fig 6: Author: Owner Subject: Highlight Date: 8/30/2018 9:53:35 AM
lower elevations, while characterized by lower-severity fire as at top of figure, can also have severe fires as in recent years with climate change, probably worth explaining in caption. Also, alpine, subalpine, and montane zones are used in text including shortly below - could be added here for more complete explanation

Lines 973-974: We have added mention of the potential influence of climate change on inferred processes from this study.

Line 332, Fig 7: Author: Owner Subject: Highlight Date: 8/30/2018 9:44:56 AM
Sp

Line 978: We have corrected this spelling mistake.

Line 345: Author: Owner Subject: Highlight Date: 8/30/2018 9:53:04 AM
Sp

Line 965: We have corrected this spelling mistake.

Line 357-358: Author: Owner Subject: Highlight Date: 8/30/2018 9:54:35 AM
too wordy, simplify

Lines 775-762: We have simplified this statement.

Line 395: Author: Owner Subject: Highlight Date: 8/30/2018 9:56:23 AM
no hyphen needed after -ly suffix, here and elsewhere

Line 854: We have deleted this sentence in preference for reorganization. We re sure not to hyphenate this phrase anywhere in the manuscript.

Lines 424-426: Author: Owner Subject: Highlight Date: 8/30/2018 10:01:53 AM
As elsewhere including in the review responses and current abstract, this should be qualified given the confounding influence of valley confinement

Lines 988-989: We have qualified this statement based on our findings related to valley confinement.

Line 485: Author: Owner Subject: Highlight Date: 8/29/2018 3:29:54 PM
Sediment

Line 1081: We have made the suggested edit.

Line 488: Author: Owner Subject: Highlight Date: 8/30/2018 10:08:22 AM
Usage inconsistent with typical definition of flood slackwater deposits e.g. Kochel and Baker 1982; gravel doesn't deposit in a low-energy environment. Along these general lines, were samples primarily from overbank sediments?

Line 1081 and 1087: We have deleted mention of slackwater deposits and included specification that we sampled overbank deposits. .

Line 494: Author: Owner Subject: Highlight Date: 8/30/2018 10:03:31 AM
sp

Line 1093: We have corrected this typo.

Line 502: Author: Owner Subject: Highlight Date: 8/29/2018 3:41:26 PM
cleaned of what? rootlets and adhering organics, presumably

Line 1102: Yes, we have clarified this statement.

Lines 504-505: Author: Owner Subject: Highlight Date: 8/29/2018 3:42:13 PM
partly redundant; weighed under a microscope?

Lines 1103-1104: We have reworded this sentence for clarity and eliminated redundancy.

Line 505: Author: Owner Subject: Highlight Date: 8/29/2018 3:53:28 PM
this needs updating and more complete explanation

Lines 1105-1106: We have updated the terminology to reflect the new analysis we conducted in our first round of edits, which include weighted means rather than pooled means, and have provided a more complete explanation of the methods pertaining to the exceedance probabilities.

Line 508: Author: Owner Subject: Highlight Date: 8/29/2018 3:55:46 PM
distributions ... were

Line 1245: We have corrected the typo as suggested.

Line 509: Author: Owner Subject: Highlight Date: 8/30/2018 8:08:19 AM
see general comments

Lines 1246-1259: Here we describe in detail the procedures and reanalysis we conducted using only the mean ages rather than all peaks in the distributions and corrected figures and text, including the methods.

Line 516: Author: Owner Subject: Highlight Date: 8/29/2018 4:02:25 PM
reach average is not pooled mean as above - clarify. Also, mean calibrated age needs to be defined - is this the weighted mean of the calibrated probability distribution for an age (e.g. Telford, R.J., Heegaard, E. and Birks, H.J., 2004. The intercept is a poor estimate of a calibrated radiocarbon age. The Holocene, 14(2), pp.296-298.)

Lines 1127-1131: We have clarified this statement as site-average of weighted mean ages and cited the reference provided when referring to weighted mean ages.

Line 517-519: Author: Owner Subject: Highlight Date: 8/29/2018 4:04:57 PM
again you need to be specific about what is meant by average radiocarbon ages here - you have just mentioned both reach averages and and mean calibrated ages

Lines 1127-1131: We have clarified this statement as site-average of weighted mean ages

Line 521: Author: Owner Subject: Highlight Date: 8/29/2018 4:05:49 PM
again, years are not units for rates

Lines 1134-1136: We have corrected this statement to refer to the given ranges as turnover times for wood.

Lines 523-524: Author: Owner Subject: Highlight Date: 8/30/2018 10:10:21 AM
assuming equal storage potential at high and low sites, correct?

Lines 1139-1140: Yes, we have added this clarification to the text.

Lines 526-527: Author: Owner Subject: Highlight Date: 8/29/2018 4:08:23 PM

charcoal can also be generated on the floodplain, where it could often represent a minimum age for the underlying sediment deposition - see general comments

Lines 1146-1212: We have provided additional text to emphasis and expand upon this point.

Line 528: Author: Owner Subject: Highlight Date: 8/29/2018 4:09:22 PM
???

Line 1145: We tried to clarify our methods based on comments from the first round of edits. However, we do understand the limitations of making this distinction this given our sampling methods so we removed mention of sub-rounded charcoal.

Line 528-529: Author: Owner Subject: Highlight Date: 8/29/2018 4:11:37 PM
not mentioned in original ms.; charcoal can be broken in augering

Line 114 and 1207-12085: We have clarified this text to communicate that we dated only larger fragments and tried to avoid dating charred rootlets.

Lines 534-538: Author: Owner Subject: Highlight Date: 8/30/2018 10:14:35 AM
I would just stick with saying that the steady-state assumption can't be fully met, given spatial differences in erosion esp. with extreme events, and I'm not convinced that the age distributions can be used to assume well-mixed reservoirs, given the small sample sizes - see general comments

Line 1219: As suggested, we have retained the text regarding spatially heterogeneous erosion and deleted additional discussion highlighted in the comments.

Lines 547-549: Author: Owner Subject: Highlight Date: 8/30/2018 10:15:46 AM
yes, appropriate to note

Lines 1146-1214: We have also expanded upon this with regard to in situ charcoal production.

Line 592: Owner Subject: Highlight Date: 8/29/2018 3:29:23 PM
not pooled

Line 1322: We have corrected this text left over from the first draft prior to reanalysis.

Below, please find the pasted comments from the annotated pdf of the supplemental material by Reviewer #1 with the appropriate reference to the line number in the prior version of the manuscript. Similar to above, we respond below each of these comments with the line number in the pdf with tracked changes where related changes were made and provide our response to each comment in blue italics.

Table S1. these need more complete explanation in a table footnote

Lines 7 – 8 and in Table 2: We have provided more thorough explanation in the caption of this table.

Line 12: sediment

Line 13: We have made the suggested edit.

Line 22: not pooled, correct?

Correct, we have corrected this text

Reviewer #2 (Remarks to the Author):

The revised manuscript is much improved. The authors clearly addressed all of the previous review comments. This work uses an analysis of radiocarbon ages from 8 watersheds in varying elevation, drainage area, degree of confinement, etc. in the Rockies, largely within Rocky Mountain National Park and an analysis of pre- and post-flood LiDAR DEMs to demonstrate that climate, fire history, geologic setting (historic Pleistocene terminal moraine) all act to control sediment residence times in floodplains of mountain rivers. The work is timely and relevant to the field and as a result will be of high interest. Transport of fine sediment through fluvial networks is a high priority topic in geomorphology for a wide variety of reasons. The text reorganization has greatly improved the clarity and readability. The more carefully worded statements regarding interpretation and discussion of assumptions are also a great improvement. The addition of the age distributions and their implications greatly add to the discussion of residence times and puts the work into context of the building literature on residence times. The modifications of the discussion of the conceptual model and its focus on elevation differences with implications for process is also a great improvement. Overall, the paper is greatly improved and I have no additional comments or changes to make. As is stands, it is accepted for publication in Nature Communications.

We appreciate prior comments by reviewer #2 and feel they helped to greatly improve the quality of this manuscript.

Sincerely,

Nicholas A. Sufin

Reviewers' comments:

Reviewer #1 (Remarks to the Author):

The authors have responded effectively to previous review comments and concerns. I have made a number of minor editorial comments and suggestions and some requests for clarification in the attached annotated PDF, none of which should require significant time to address. One concern is that averages of the radiocarbon ages at each site are sometimes described as weighted means - are they really weighted in some way? My understanding is that only the individual radiocarbon ages are expressed as the weighted mean of the calibrated probability distribution (see comments on related highlighted text). Overall, with very minor revision, this paper will be ready for publication in Nature Communications, and I look forward to seeing it in print.

Grant Meyer

REVIEWERS' COMMENTS:

The authors have responded effectively to previous review comments and concerns. I have made a number of minor editorial comments and suggestions and some requests for clarification in the attached annotated PDF, none of which should require significant time to address. One concern is that averages of the radiocarbon ages at each site are sometimes described as weighted means - are they really weighted in some way? My understanding is that only the individual radiocarbon ages are expressed as the weighted mean of the calibrated probability distribution (see comments on related highlighted text). Overall, with very minor revision, this paper will be ready for publication in Nature Communications, and I look forward to seeing it in print.
Grant Meyer

Again, we greatly appreciate the time and attention Grant Meyer have devoted to commenting and helping to improve our manuscript. We have specifically addressed comments regarding weighted mean ages and made to clarify this in the entirety of the text so that weighted means is correctly used only when we refer to weighted means of individual ages. We refer to site-average means among all samples at each study site as a proxy for residence time, which does not involve weighted averages.

Lines 20-22: 2 separate topics, reword in separate sentences, e.g. start following sentence with "Fifty-two 14C ages shat that ..."

Lines 20-23: We have made the suggested edit and rewording.

Lines 19-22:

Line 40: where

Line 48: We have made the suggested edit

Line 89:

Lines 52-54: sentence fits better following line 39; this paragraph is about wildfire effects

Lines 45-47: We have made the suggested move for this sentence.

Lines 105-107:

Line 80: First explain fully that "Radiocarbon ages were obtained for 2 to 15 large charcoal pieces (>1 cm³) at each of 8 study sites, and the weighted mean of the calibrated probability distribution was calculated for each age, providing proxies for floodplain sediment ages". Then explain that "Average ages were calculated as the arithmetic mean of ages at that site."

Lines 100-104: We have made the suggested restructuring and clarification of these sentences.

Lines 225-229:

Line 82: sites

Line 104: We have corrected this typo

Lines 223:

Lines 89-90: could use a little more explanation

Lines 110-113: We have added additional explanation to this statement and added clarification to the following sentence to avoid confusion regarding the subject of the sentence.

Lines 115-118:

Lines 108-110: wording indicates you are comparing the montane zone to another zone – but you haven't mentioned the subalpine zone yet at all. Define each briefly by elevation first, then your comparison will make sense

Lines 139-140: We have added mention of both zones sooner in the paragraph as suggested.

Lines 108-211:

Line 116: on average

Line 154: We have made the suggested edit

Lines 115-118:

Lines 129-130: was used to

Lines 178-179: We have made the suggested edit

Lines 814:

Line 135: or with beaver evidence?

Line 184-185: We have made the suggested edit

Lines 1734-1735:

Line 140: this should be defined very briefly along with the montane and subalpine zones, above

Line 139-141: We have very briefly added mention of and the elevation range for the alpine zone in the first mention of ecozones.

Line 158-163: We provide here limited details about the alpine zone

Lines 817-823:

Line 176: relatively

Line 228: We have make the suggested deletion

Line 1042:

Line 189: The hydroclimatic elevation threshold

Line 241: We have made the suggested edit

Line 991:

Line 208: accelerator

Line 261: We have corrected this typo

We have deleted this sentence because of redundancy and the correct spelling appears on line 1116

Line 209: no –

Line 262: We have corrected this typo

Line 1118:

Line 210: yes – site averages are proxies for sediment residence time; individual ages are proxies for sediment age as in comment line 80

Line 1119: We have clarified this in the previous comment on line 80 and are sure to make the distinction

Line 230: tacked on at end – reword. Again you are comparing, so include Blong and Gillespie study area size and wood type.

Lines 285-287: We have added details regarding this comparison including the hard wood type and drainage area size from the cited study Blong and Gillespie

Line 759-762: We have deleted the original sentence to reduce redundancy, which is substituted by this sentence here.

Line 233 compounded?

Line 289: This makes more sense and we have made the suggested edit

Line 765:

Line 242: The site mean is the average of the weighted means of the calibrated probability distributions for the ages at that site, yes? Word as "site average of the weighted mean 14C ages"

Line 305: The reviewer question/comment is correct and we have made the suggested edit.

Line 1794:

Line 251: yes – hyphenate where this is a compound adjective

Duly noted

Line 263: citation style

Line 328: we have corrected improper citation format

Line 744:

Line 264: add this

Line 329: We have correctly added this citation

Line 742:

Line 277: delete “follow an exponential relationship”

Line 346: We have made the suggested deletion

Line 353:

Line 278: no

Line 347: We have corrected this typo

Line 354: We have deleted this sentence.

Line 333: fire-scarred

Line 403: We have made the appropriate edit

Line 572:

Lines 352-353: summed probabilities of radiocarbon ages after 1500 CE compared with tree-ring dated wildfire ages (vertical lines, color-coded to match wildfires mapped in Figure 1B).

Lines 423-425: We have made the suggested edit.

Line 1759-1761:

Lines 366-367: When specifically in the century does Woodhouse ID drought?

Lines 442-443: We have indicated more precisely that location and periods of drought summarized by Woodhouse and Overpeck as they relate to our data.

Line 1613-1614:

Line 373: use ; instead of because – the latter part of this sentence doesn't explain why, it just describes the same thing as the first part in different words

Line 449: We have made the suggested edit.

Line 620:

Lines 430-431: is a significant predictor

Line 511: We have corrected this typo as suggested

Line 711-712:

Lines 450-451: debris-flow generation should be in here somewhere, which can be runoff-related as well as from slope failures

Line 534: We have made the suggested edit

Line 838:

Line 459: reword – results don't examine

Lines 544-546: We have corrected this sentence with more precise language .

Line 860:

Line 463: stabilizes is misspelled in Fig 6A; 6C should explicitly include increased postfire debris flow activity and sed transport from slopes

Response to reviewer comments for NCOMMS-18-03847C

We have updated the figure, correcting the typo and including more explicit language regarding debris flows and potential changes with a changing climate.

Line 466: severity

As above, we have made the suggested edit, which is included in the figure, but not the caption.

Line 495: delete and delete

Lines 587: we have made the suggested edits.

Line 416:

Line 500: , showing that

Line 592: we have made the suggested edit

Line 921

Lines 506-508: true enough, but this isn't addressed in your study

Line 598: We have deleted this statement

Lines 584: ? implies pencil and paper...

Line 681: We have deleted the word, "manually"

Line 1132:

Line 588: it's not really a weighted average, though, is it?

Line 687: That is corrected, we should have said site-average of weighted mean ages. However, we do not use that terminology consistently in the manuscript, so we have deleted the word "weighted" here.

We have deleted this sentence to reduce redundancy in the final version

Line 601: though the probability of softwood charcoal being reduced to very fine pieces that would not be sampled also increases downstream – worth mentioning

Lines 701-704: We have made the suggested edit to include a statement about the increased likelihood of charcoal pieces breaking down with transport downstream and the decreased chance that they would be sampled at lower elevations

Lines 770-773:

Line 607: fragments

Line 715: We have made the suggested edit.

We have deleted this sentence to reduce redundancy in the final version

Line 608: were rootlets removed from inside charcoal pieces before dating?

Line 715-716: Yes, we removed fine rootlets under a low power microscope and have edited the text to state such here.

Lines 114-115: